# Identification of a prefrontal cortex-to-amygdala pathway for chronic stress-induced anxiety

Wei-Zhu Liu[1,2,5], Wen-Hua Zhang[1,5], Zhi-Heng Zheng[1], Jia-Xin Zou[1], Xiao-Xuan Liu[1], Shou-He Huang[1], Wen-Jie You[1], Ye He[1], Jun-Yu Zhang[1], Xiao-Dong Wang[3] & Bing-Xing Pan [1,2,4 ✉]

Dysregulated prefrontal control over amygdala is engaged in the pathogenesis of psychiatric diseases including depression and anxiety disorders. Here we show that, in a rodent anxiety model induced by chronic restraint stress (CRS), the dysregulation occurs in basolateral amygdala projection neurons receiving mono-directional inputs from dorsomedial prefrontal cortex (dmPFC→BLA PNs) rather than those reciprocally connected with dmPFC (dmPFC↔BLA PNs). Specifically, CRS shifts the dmPFC-driven excitatory-inhibitory balance towards excitation in the former, but not latter population. Such specificity is preferential to connections made by dmPFC, caused by enhanced presynaptic glutamate release, and highly correlated with the increased anxiety-like behavior in stressed mice. Importantly, low-frequency optogenetic stimulation of dmPFC afferents in BLA normalizes the enhanced prefrontal glutamate release onto dmPFC→BLA PNs and lastingly attenuates CRS-induced increase of anxiety-like behavior. Our findings thus reveal a target cell-based dysregulation of mPFC-to-amygdala transmission for stress-induced anxiety.

---

[1] Laboratory of Fear and Anxiety Disorders, Institutes of Life Science, Nanchang University, 330031 Nanchang, China. [2] Department of Biological Science, School of Life Science, Nanchang University, 330031 Nanchang, China. [3] Department of Neurobiology, Key Laboratory of Medical Neurobiology of Ministry of Health of China, Zhejiang Province Key Laboratory of Neurobiology, Zhejiang University School of Medicine, 310058 Hangzhou, China. [4] Department of Ophthalmology, the 2nd Affiliated Hospital, Medical School of Nanchang University, 330031 Nanchang, China. [5] These authors contributed equally: Wei-Zhu Liu, Wen-Hua Zhang. ✉email: panbingxing@ncu.edu.cn

In the brain, the medial prefrontal cortex (mPFC) and amygdala are extensively interconnected and work in concert to tune the expression of emotions, such as fear and anxiety[1–4]. Under physiological conditions, the mPFC exerts inhibitory top–down control over amygdala activity, limiting its output and thus preventing inappropriate emotion expression[5–8]. However, under adverse conditions such as prolonged exposure to inescapable stress that precipitates the development of psychiatric conditions including anxiety disorders and depression, the prefrontal control becomes defective, resulting in aberrant amygdala activation and deficits in emotion and behavior[9–11]. The defect has been frequently reported in both human psychiatric patients and animal models and recognized as one of the core neurobiological features across stress-associated neuropsychiatric disorders[12–17].

Mounting anatomical evidence has shown that the amygdala-projecting mPFC neurons send most of their fibers to the basolateral amygdala (BLA) as opposed to the lateral or central subregion[18,19]. The BLA is composed of excitatory projection neurons (PNs, 80–90%) and inhibitory interneurons (10–20%)[20], with the PNs mediating BLA's communication with other brain regions under the fine-tuning of interneurons. Both neuron groups are the targets of mPFC[21,22]. For PNs, although they are spatially intermingled in BLA, recent studies have identified considerable between-PN heterogeneity in terms of their gene expression, connections with extra-amygdala regions, and roles in amygdala-associated tasks[23–27]. Given the crucial role of defective mPFC-to-amygdala communication for the pathogenesis of stress-related neuropsychiatric illness, two questions arise. First, during exposure to chronic or extreme stress, what specific changes occur in mPFC connections to the diverse BLA PNs? Second, how do the changes in these connections contribute to the emotional and behavioral adversity by stress?

To address these issues, we employ a rodent anxiety model induced by chronic restraint stress (CRS) exposure. The BLA PNs are classified into two clusters based on their differential connection with dorsal mPFC (dmPFC), with one being reciprocally connected with dmPFC (dmPFC↔BLA PNs) and the other only receiving mono-directional dmPFC afferents (dmPFC→BLA PNs). We have recently observed different adaptation of the two populations to CRS in terms of their synaptic architecture and neuronal excitability[28,29]. Here we explore the regulations of CRS on dmPFC transmission to these distinct PNs and their specific roles in CRS-induced anxiety. We observe that CRS selectively shifts the dmPFC-driven excitatory–inhibitory (E–I) balance toward excitation in dmPFC→BLA but not in dmPFC↔BLA PNs as a consequence of selective increase of presynaptic glutamate release onto the former population. Importantly, the increased prefrontal glutamate release onto dmPFC→BLA PNs well correlates with the increased anxiety-like behavior in stressed mice. Using low-frequency optogenetic stimulation to normalize the augmented prefrontal glutamate release suffices to counteract stress-induced increase of anxiety-like behavior. Thus we identify target cell connectivity-based dysregulation of dmPFC-to-BLA pathway for stress-induced increase of anxiety. Targeting the altered communication in this pathway may be of translational value for treatment of stress-related psychiatric disorders.

## Results

**CRS shifts dmPFC-driven E–I balance in BLA**. We first examined the regulation of dmPFC-to-BLA transmission by CRS. For this, we stereotaxically injected an adeno-associated viral vector carrying channelrhodopsin 2 (AAV-ChR2) tagged with enhanced green fluorescent protein (eGFP) under control of

CaMKII promoter into bilateral dmPFC of mice (Fig. 1a, b). In line with earlier finding[21], we observed dense ChR2-expressing dmPFC fibers in BLA but not in LA or central amygdala (Fig. 1b). Two weeks after injection, mice were subjected to either a 2-h restraint stress (CRS) or 5-min gentle handling (unstressed control) per day for a continuum of 10 days. Twenty-four hours after the last episode of stress or handling, the slices of amygdala were cut and the responses of BLA PNs to dmPFC inputs were examined by using whole-cell patch recordings (Fig. 1c, d). Optogenetic activation of the dmPFC fibers evoked robust biphasic responses in BLA PNs (held at −30 mV), with an initial inward excitatory postsynaptic current (EPSC) being followed by an outward inhibitory postsynaptic current (IPSC) (Fig. 1e). The EPSCs are presumably monosynaptic, on considering their short onset latency (~4 ms), their complete blockade by tetrodotoxin (TTX), a sodium channel blocker, and subsequent partial rescue by co-application of 4-AP, a potassium channel blocker (Supplementary Fig. 1a–c). The IPSCs had onset latency twice that of EPSCs and were completely blocked by either the glutamatergic receptor antagonists DL-AP5 and CNQX or GABA$_A$ receptor antagonist picrotoxin (Supplementary Fig. 1d–f), indicating that they are disynaptic and due to the activation of local inhibitory network following dmPFC inputs.

By step-increasing the intensity of light pulse, we constructed the input–output relationship of EPSCs and IPSCs in BLA PNs from CRS and unstressed mice. Relative to the unstressed controls, the CRS mice had higher efficacy of EPSCs but similar efficacy of IPSCs (Fig. 1e–g), resulting in a significantly lower ratio of IPSCs over EPSCs (Fig. 1h). These results indicate that CRS shifts dmPFC-driven E–I balance toward excitation in BLA PNs via potentiating the monosynaptic glutamatergic transmission.

To test whether the shift by CRS also occurred in other inputs to BLA, we repeated the above comparisons in ventral mPFC- (vmPFC) and ventral hippocampus (vHPC)-to-BLA pathways. ChR2-carrying AAV vectors were injected to vmPFC or vHPC to allow ChR2 expression in their terminals within BLA (Fig. 1i–v). Similar to that of dmPFC inputs, light activation of inputs from the two regions also evoked biphasic responses in BLA PNs (Fig. 1l, s). However, CRS had little influence on the amplitudes of vmPFC- (Fig. 1l–o) or vHPC-evoked (Fig. 1s–v) EPSCs and IPSCs as well as their ratios, suggesting that inputs from the two regions to BLA are more resilient against CRS than those from dmPFC. Thus CRS preferentially disrupts dmPFC-driven E–I balance in BLA.

**CRS augments dmPFC transmission onto dmPFC→BLA PNs**. The above results revealed that CRS induced shift of dmPFC-driven E–I balance to excitation in BLA PNs as a whole. Given the considerable anatomical and functional heterogeneity across individual BLA PNs[24,30], we next explored how CRS affected dmPFC transmission to these distinct neurons. The BLA PNs were clustered into dmPFC→BLA and dmPFC↔BLA PNs based on their connectivity with dmPFC. To differentiate the two populations, we co-injected the red fluorescent Retrobeads into dmPFC with ChR2-carrying AAV into dmPFC (Fig. 2a, b). Since virtually all BLA PNs are innervated by dmPFC[21], we designated the beads-labeled BLA PNs as putative dmPFC↔BLA PNs and those unlabeled as putative dmPFC→BLA ones.

The CRS effects on the input–output relationship of dmPFC-evoked EPSCs and IPSCs were then investigated in the two populations (Fig. 2c). While robustly enhancing the efficacy of EPSCs in dmPFC→BLA PNs, CRS did not affect that in dmPFC↔BLA PNs (Fig. 2d–g). And, in keeping with its negligible effect on GABAergic transmission in BLA PNs as a whole (Fig. 1g),

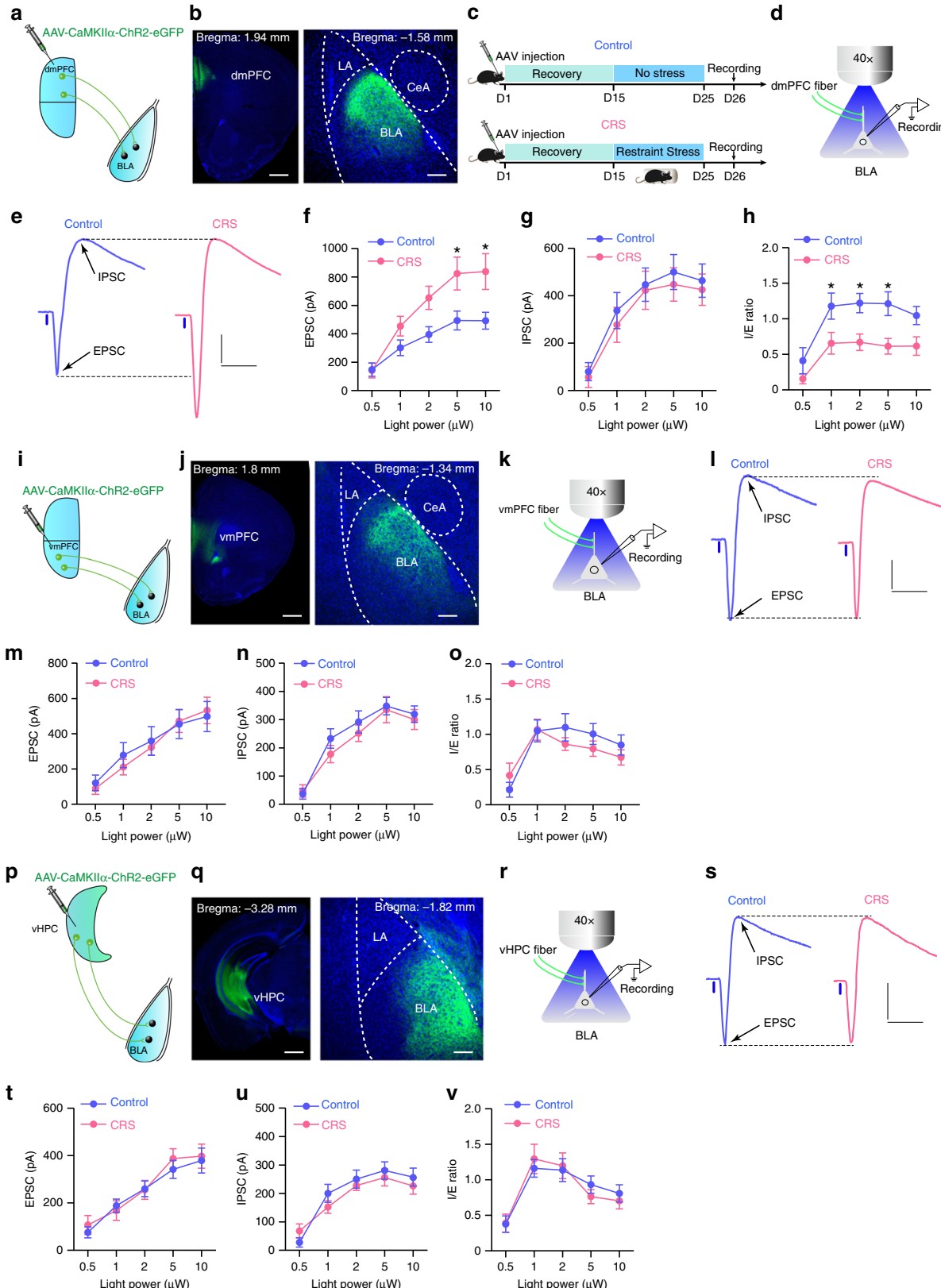

CRS altered the efficacy of IPSCs in neither population (Fig. 2e–g). Consequently, the ratio of IPSCs over EPSCs was markedly decreased in dmPFC→BLA PNs following CRS but remained unaltered in dmPFC↔BLA PNs (Fig. 2h, i). Thus it appears that CRS dysregulates dmPFC-to-BLA transmission in a manner heavily dependent on the connectivity of target BLA PNs, resulting

in E–I imbalance in dmPFC→BLA PNs rather than their dmPFC↔BLA neighbors. In line with the undisturbed vmPFC- or vHPC-to-BLA transmission following CRS (Fig. 1), the evoked EPSCs and IPSCs by inputs from the two regions as well as their ratios were altered in neither population (Supplementary Figs. 2 and 3).

**Fig. 1 Chronic stress shifts dmPFC but not vmPFC or vHPC-driven excitatory–inhibitory balance to excitation in BLA PNs. a** Schematic showing injection of ChR2-carrying adeno-associated virus (AAV) vectors into dmPFC for optogenetic activation of dmPFC inputs to BLA. **b** Representative images showing the injection site in dmPFC (left) and the dmPFC inputs in BLA (right). DAPI staining was used to outline the brain regions of BLA. Scale bar: 500 (left) and 100 (right) μm. **c** Experimental procedures for control mice and mice subjected to chronic restraint stress (CRS). **d** Schematic showing recording of BLA PN responses to optogenetic stimulation of dmPFC inputs. **e** Representative traces showing the evoked EPSCs/IPSCs in BLA PNs of control and CRS mice following light activation of dmPFC inputs. Cells were held at −30 mV. Scale bar = 10 ms, 200 pA. **f–h** Summary plots of the EPSCs (**f**), IPSCs (**g**), and the IPSC/EPSC (I/E) ratio (**h**) in **e** with increasing light intensities. Control, $n = 12$ neurons/4 mice; CRS, $n = 13$ neurons/4 mice. **i, j** Same as in **a**, **b** except that the AAV vectors were injected into vmPFC. **k** Schematic showing recording of BLA PN responses to optogenetic activation of vmPFC inputs. **l** Representative traces showing the evoked EPSCs/IPSCs in BLA PNs of control and CRS mice following light activation of vHPC inputs. Cells were held at −30 mV. Scale bar = 10 ms, 200 pA. **m–o** Summary plots of the EPSCs (**m**), IPSCs (**n**), and I/E ratio (**o**) in **l** with increasing light intensity. Control, $n = 12$ neurons/4 mice; CRS, $n = 14$ neurons/5 mice. **p, q**, Same as in **a**, **b** except that the AAV vectors were injected into vHPC. **r** Schematic showing recording of BLA PN responses to optogenetic activation of vHPC inputs. **s** Representative traces showing the evoked EPSCs/IPSCs in BLA PNs of control and CRS mice following light activation of vHPC inputs. Cells were held at −30 mV. Scale bar = 10 ms, 200 pA. **t–v** Summary plots of the EPSCs (**t**), IPSCs (**u**), and the I/E ratio (**v**) in **s** with increasing light intensity. Control, $n = 14$ neurons/5 mice; CRS, $n = 15$ neurons/5 mice. Data are presented as mean ± SEM. Statistics are shown in Supplementary Table 1. *$p < 0.05$. Source data are provided as a Source Data file.

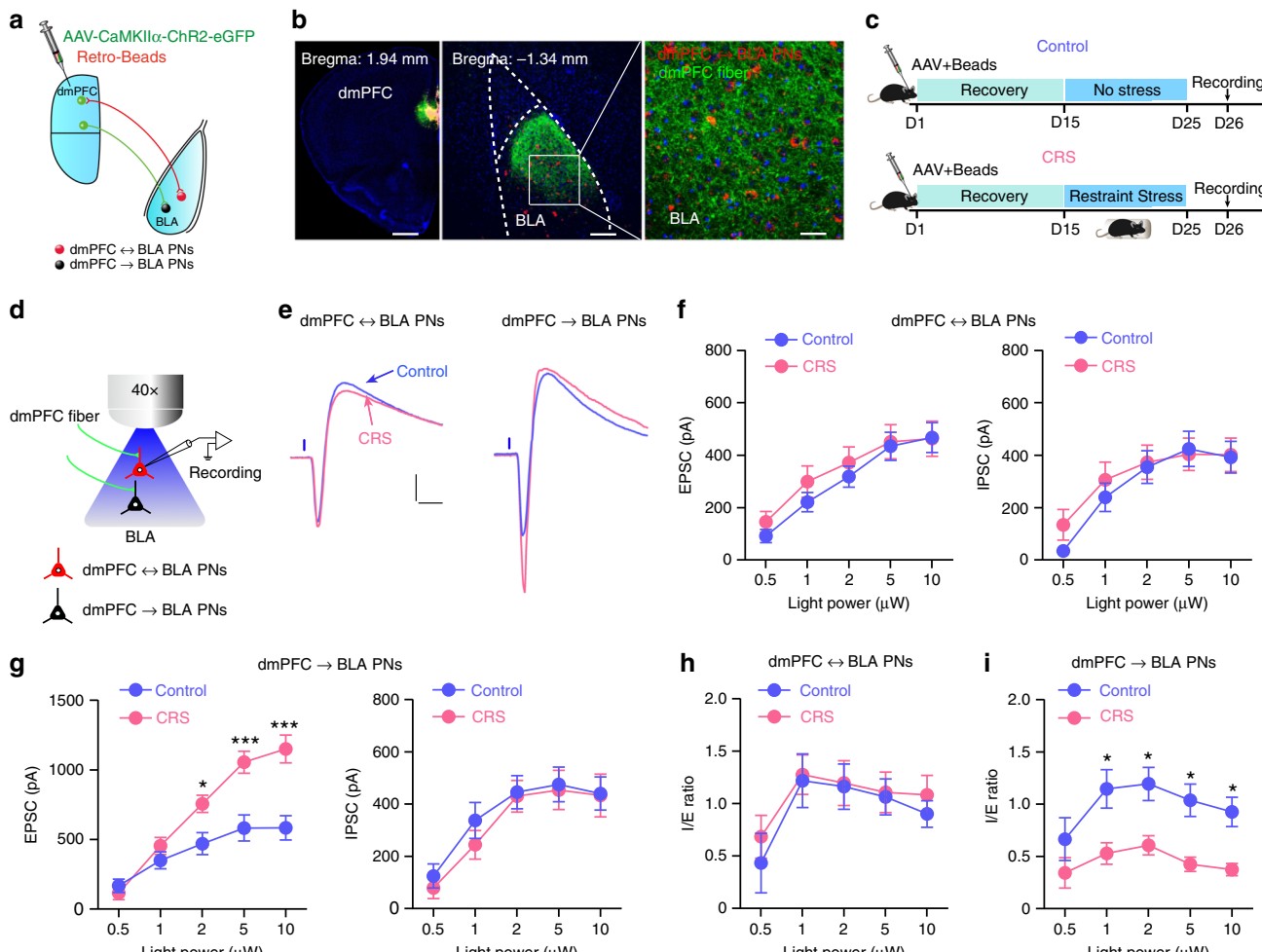

**Fig. 2 Chronic stress selectively augments dmPFC-evoked glutamatergic transmission onto dmPFC→BLA PNs. a** Schematic showing co-injection of ChR2-carrying AAV vectors and red fluorescent Retrobeads into dmPFC. Retrobeads were used to differentiate the putative dmPFC→BLA and dmPFC↔BLA PNs in BLA. **b** Representative images showing the injection site in dmPFC (left) and red Retrobeads-labeled dmPFC↔BLA PNs and dmPFC inputs in BLA (middle). The expanded square was shown on right. Scale bar: 500, 100, and 20 μm (left to right). **c** Schematic showing the experimental procedures for control mice and mice subject to chronic restraint stress (CRS). **d** Schematic showing recording of the postsynaptic responses in dmPFC→BLA or dmPFC↔BLA PNs to optogenetic activation of dmPFC inputs. **e** Representative traces showing the biphasic EPSCs/IPSCs in dmPFC↔BLA and dmPFC→BLA PNs following optogenetic activation of dmPFC inputs. Cells were held at −30 mV. Scale bar = 10 ms, 200 pA. **f** Summary plots of the EPSCs (left) and IPSCs (right) in dmPFC↔BLA PNs with increasing light intensity. Control, $n = 14$ neurons/4 mice; CRS, $n = 15$ neurons/4 mice. **g** Same as in **f** except that the data were from dmPFC→BLA PNs. Control, $n = 15$ neurons/5 mice; CRS, $n = 13$ neurons/4 mice. **h** Summary plots of the I/E ratio in dmPFC↔BLA PNs. Same sample size as in **f**. **i** Same as in **h** except that the data were from dmPFC→BLA PNs. Same sample size as in **g**. Data are presented as mean ± SEM. Statistics are shown in Supplementary Table 1. *$p < 0.05$, ***$p < 0.001$. Source data are provided as a Source Data file.

To further confirm the target cell-based regulation of CRS on dmPFC-to-BLA transmission, we performed two additional experiments. First, we isolated the EPSCs by holding the cells at $-70\,mV$ in the presence of picrotoxin and CGP52432 to block $GABA_A$ and $GABA_B$ receptor-mediated currents, respectively. The augmentation of dmPFC-evoked EPSCs by CRS was also observed in dmPFC→BLA but not in dmPFC↔BLA PNs (Supplementary Fig. 4a–d). Second, we compared the impacts of CRS on dmPFC-evoked, N-methyl-D-aspartate (NMDA) receptor (NMDAR)-mediated eEPSCs in the two populations. We found that it selectively enhanced the currents in the former but not latter population (Supplementary Fig. 4e–h). Collectively, these findings strongly suggest CRS-mediated dysregulation of dmPFC-to-BLA communication preferentially occurs in dmPFC→BLA PNs rather than their dmPFC↔BLA neighbors.

**Chronic corticosterone (CORT) usage recapitulates effects of CRS.** The brain dysfunction by diverse stressors is commonly associated with the increased secretion of glucocorticoid hormone (CORT in rodents). Chronic CORT treatment was found to recapitulate multiple cellular and behavioral phenotypes by prolonged stress[31–34]. To test whether chronic CORT administration was sufficient to cause target cell-dependent, dmPFC-driven E–I imbalance in BLA, mice were allowed to drink CORT-containing water (0.1 mg/ml) or vehicle solution for 10 consecutive days. Relative to the vehicle controls, the CORT-treated mice displayed stronger dmPFC-evoked EPSCs but intact IPSCs in their BLA→dmPFC PNs, yielding a lower I–E ratio in this population. By contrast, these differences were not observed in dmPFC↔BLA PNs (Supplementary Fig. 5). To control the consumed volume of CORT among animals, we repeated the experiment by using commercial slow-release CORT pellets or placebo[35]. Similarly, CORT pellet-treated mice also displayed stronger dmPFC-evoked EPSCs, intact IPSCs, and lower I–E ratio in their BLA→dmPFC PNs relative to the placebo-treated ones. No such differences were observed in dmPFC↔BLA PNs (Supplementary Fig. 6). Taken together, these results strongly suggest that chronic CORT administration is sufficient to recapitulate the effects of CRS on dmPFC-to-BLA pathways and the dysregulated dmPFC inputs to dmPFC→BLA PNs may represent a common pathology among mice experiencing prolonged exposure to different stressors.

**CRS increases glutamate release onto dmPFC→BLA PNs.** We next explored the synaptic mechanisms through which CRS selectively potentiates the prefrontal glutamatergic transmission to dmPFC→BLA PNs. The potentiation may be caused by either increased presynaptic glutamate release or augmented postsynaptic response to glutamate, or both. We began to evaluate the probability of presynaptic glutamate release (Pr) in dmPFC-to-BLA connections via two approaches. First, by delivering two consecutive light pulses with varying intervals to excite the dmPFC afferents, we looked at the paired-pulse ratio (PPR) of eEPSCs, which was known to be inversely correlated with the Pr of glutamate. The PPR in dmPFC→BLA PNs was far lower in CRS mice than the unstressed controls but comparable between the dmPFC↔BLA PNs of the two groups, reflecting a selective increase of Pr in dmPFC inputs to dmPFC→BLA PNs by CRS (Fig. 3a–d). Second, we directly compared the Pr by measuring the decay of NMDAR-mediated currents upon repetitive stimuli of dmPFC inputs in the presence of MK-801, a non-competitive NMDAR antagonist. We observed that the decay constant in dmPFC→BLA PNs was far lower in CRS mice than their unstressed controls but was similar between the dmPFC↔BLA PNs of the two groups (Fig. 3e–h), further confirming the

selective increase of Pr in dmPFC synapses terminating on dmPFC→BLA PNs.

We also investigated the possible postsynaptic changes by CRS. First, by comparing the ratio of dmPFC-evoked, AMPA receptor-mediated EPSCs over those by NMDA receptor in BLA PNs, a signature of postsynaptic plasticity in neurons, we found that it altered the ratio in neither population (Fig. 4a–d). Second, by constructing the current–voltage (I/V) curves of AMPA and NMDA receptor currents, we observed that the curves for both remained intact in the two populations subsequent to CRS (Fig. 4e–j). Finally, by replacing extracellular $Ca^{2+}$ with equal concentrations of $Sr^{2+}$, we quantified the size of quantal responses in dmPFC transmission to BLA PNs and found that CRS had effect in neither population (Fig. 4k–p). Taken together, the above findings suggest that CRS does not affect the postsynaptic function of dmPFC synapses onto BLA PNs. Consistent with the increased glutamate release onto dmPFC→BLA but not dmPFC↔BLA PNs following CRS (Fig. 3), it elevated the frequency of quantal response in the synapses onto the former but not the latter population (Fig. 4l, o).

**Increased glutamate release correlates with anxiety in CRS mice.** Our above findings have thus far demonstrated that CRS selectively shifts dmPFC-driven E/I imbalance in dmPFC→BLA but not dmPFC↔BLA PNs as a result of increased presynaptic glutamate release onto the former population. Given the prominent role of altered prefrontal control of amygdala in the pathogenesis of stress-related neuropsychiatric diseases including anxiety disorders[36,37], we next asked whether the preferential increase of prefrontal glutamate release onto dmPFC→BLA PNs might contribute to the behavioral and emotional deficits by CRS. To this end, we first tested the correlation for both control and CRS mice between their anxiety-like behavior and the Pr in dmPFC projections to the two PN populations. The mouse anxiety-like behaviors were tested with elevated plus maze (EPM) and open field test (OFT) (Fig. 5a–c) and the Pr was measured with PPR 4 h post the behavioral test.

In keeping with the earlier findings[28,38], CRS mice showed more prominent anxiety-like behavior than the unstressed controls. In EPM, they spent shorter time in and had fewer entry numbers to the open arms (Fig. 5d, e). In OFT, they spent shorter time in the center region and had similar travel distance to the unstressed controls (Fig. 5f, g). Subsequent neuronal recordings replicated the decrease of PPR in dmPFC synapses targeting BLA→dmPFC PNs (Supplementary Fig. 7). For dmPFC↔BLA PNs, no correlations were observed between the PPR and any of the three behavioral parameters indicating anxiety-like behavior in either control or CRS mice (Fig. 5h–j). By contrast, for dmPFC→BLA PNs, there were positive and significant correlations between the PPR and all these parameters in CRS but not in unstressed mice (Fig. 5k–m), providing evidence linking the enhanced prefrontal glutamate release onto dmPFC→BLA PNs to the increased anxiety-like behavior in CRS mice.

**Reversal of increased prefrontal glutamate release by CRS.** Did the heightened prefrontal glutamate release onto dmPFC→BLA PNs drive the increase of anxiety-like behavior in CRS mice? To answer this question, we first attempted to search for approaches to attenuate the CRS influences on dmPFC-to-BLA transmission and behavior. Earlier studies have successfully used low frequency of optogenetic stimulation to suppress the increased glutamate release in mPFC output terminals by cocaine exposure[39,40]. We here optimized the stimulation protocol (1 Hz × 3 or 10 min with the pulse duration of 2 ms) in a hope to find protocols capable of reversing the increased prefrontal glutamate release onto

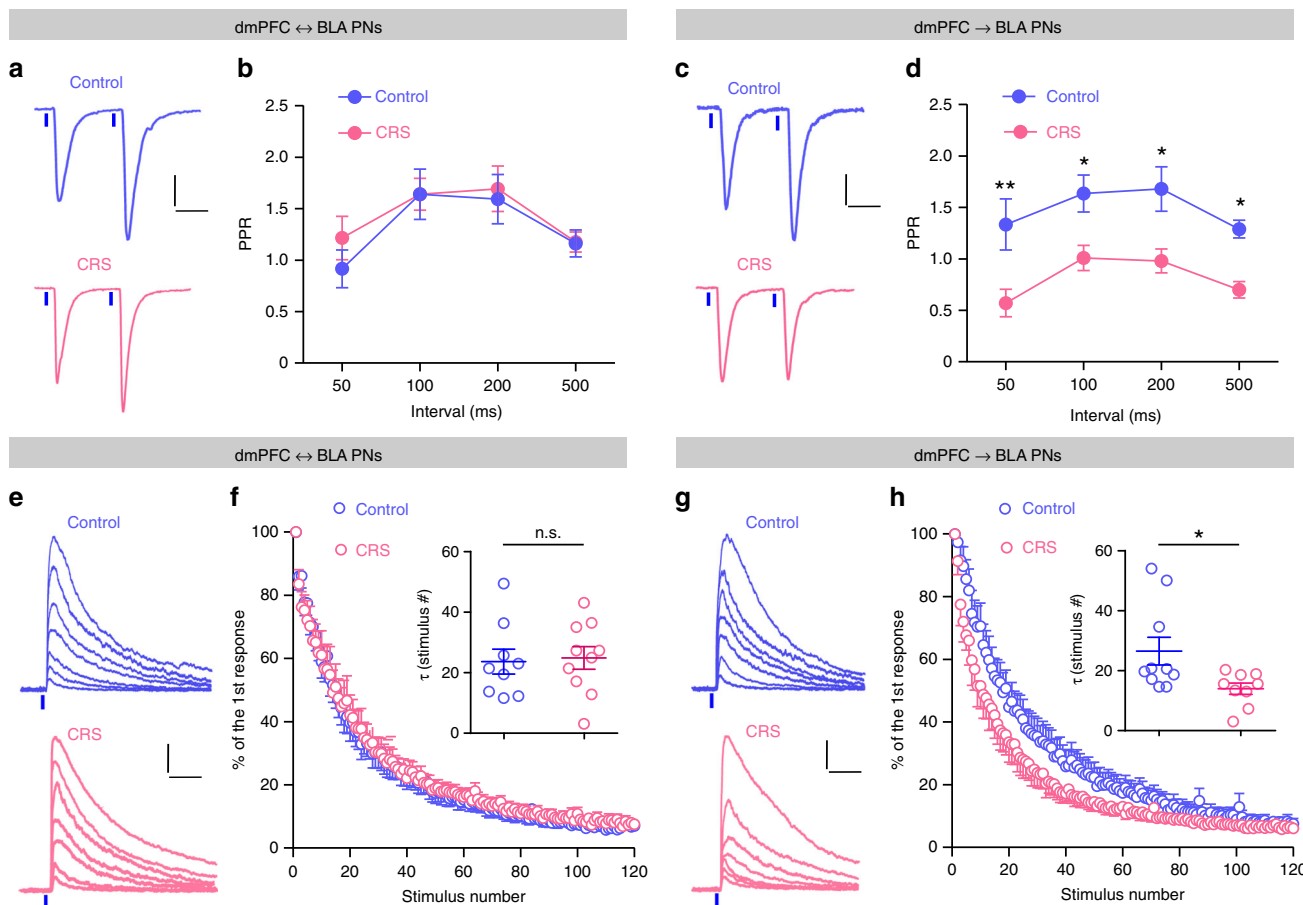

**Fig. 3 Chronic stress selectively increases prefrontal cortical glutamate release onto dmPFC→BLA PNs. a** Representative traces showing EPSCs in dmPFC↔BLA PNs upon paired light stimuli of dmPFC inputs (separated by 100 ms). Scale bar = 50 ms, 50 pA. **b** Summary plots of paired pulse ratio (PPR) in dmPFC↔BLA PNs. Control, $n = 13$ neurons/5 mice; CRS, $n = 14$ neurons/5 mice. **c** Same as in **a** except that the data were from dmPFC→BLA PNs. Scale bar = 50 ms, 50 pA. **d** Summary plots of PPR in dmPFC→BLA PNs. Control, $n = 14$ neurons/5 mice; CRS, $n = 13$ neurons/5 mice. **e** Representative traces showing the NMDA current in dmPFC↔BLA PNs in response to the 1st, 10th, 20th, 40th, 60th, 90th, and 120th photostimulation during MK-801 perfusion. Scale bar = 50 ms, 100 pA. **f** The progressive blockade of NMDA currents by MK-801 during light activation of dmPFC inputs. Each point represented averaged currents normalized to the first one. Inset showing comparison of the averaged $\tau$ values. Control, $n = 9$ neurons/4 mice; CRS, $n = 10$ neurons/4 mice. **g** Same as in **e** except that the data were from dmPFC→BLA PNs. Scale bar = 50 ms, 100 pA. **h** Same as in **f** except that the data were from dmPFC→BLA PNs. Control, $n = 10$ neurons/4 mice; CRS, $n = 9$ neurons/4 mice. Data are presented as mean ± SEM. Statistics are shown in Supplementary Table 1. n.s., not significant; *$p < 0.05$, **$p < 0.01$. Source data are provided as a Source Data file.

dmPFC→BLA PNs but unaffecting that onto dmPFC↔BLA PNs in CRS mice (Fig. 6a).

PPR was used to monitor presynaptic glutamate release in the dmPFC-to-BLA connections of CRS mice. As shown in Fig. 6b, c, 3-min ex vivo light stimulation (LS) failed to affect the PPR in both connection types. However, when the LS duration was extended to 10 min, it caused a persistent and robust increase of PPR in dmPFC synapses terminating on dmPFC→BLA but not on dmPFC↔BLA PNs (Fig. 6d, e). These results demonstrated that 10- but not 3-min LS of dmPFC inputs selectively downregulated the already increased presynaptic glutamate release onto dmPFC→BLA PNs in CRS mice.

We next tested whether in vivo LS also had similar effects. An optical fiber was implanted onto the amygdala of CRS mice immediately after injection of AAV vector containing ChR2 in dmPFC (Fig. 6f). One day after the last restraint stress, mice were subjected to in vivo LS in their BLA (3 or 10 min) or unstimulated. The PPR in dmPFC-to-BLA connections were recorded 4 h later. As shown in Fig. 6g, h, the PPRs in synapses targeting dmPFC↔BLA PNs were indistinguishable among the unstimulated mice and mice experiencing LS for either 3 or 10 min, indicating inability of LS to alter prefrontal glutamate

release onto this population. By contrast, in synapses targeting BLA→dmPFC PNs (Fig. 6g, i), the PPR was far higher in mice receiving 10-min LS than those stimulated for only 3 min or non-stimulated, suggesting 10-min in vivo LS suppresses the CRS-induced increase of prefrontal glutamate release onto this population. No effects of LS were found on the AMPA/NMDA ratio in the two types of synapses (Fig. 6j–l), arguing against a role for LS in regulating postsynaptic function.

Altogether, the above results demonstrate that 10-min LS effectively reversed the increased prefrontal glutamate release onto dmPFC→BLA PNs in CRS mice but did not affect that onto dmPFC↔BLA PNs.

**Optogenetic attenuation of increased anxiety in CRS mice**. We next investigated whether low-frequency stimulation of dmPFC inputs in BLA could attenuate the increased anxiety-like behavior in CRS mice. The experimental procedures are shown in Fig. 7a. As shown in Fig. 7b–e, 4 h after 10-min LS, the mice spent more time in the open arm during EPM and exhibited a clear, although insignificant, tendency to have more entry in this arm. Moreover, the LS-treated mice also spent more time in the center region

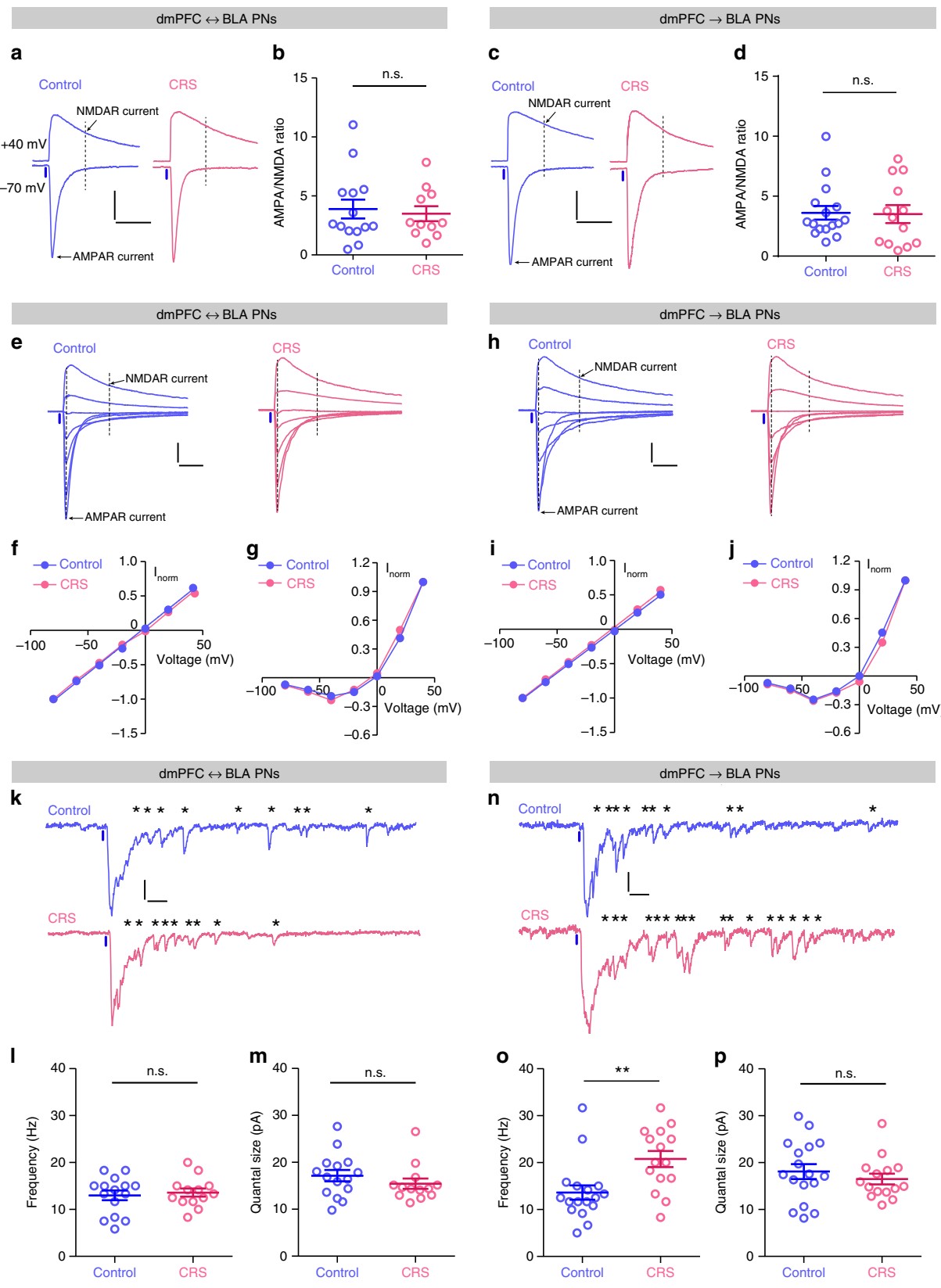

during OFT and the effect persisted for at least 24 h post-LS (Fig. 7f), suggesting lasting influence of LS. By contrast, 3-min LS failed to affect the mice's behavior in EPM and OFT (Fig. 7g–j). Both 3- and 10-min LS did not alter the distance mice traveled in OFT and their movement speed (Supplementary Fig. 8a–f). These

results suggest that 10-min LS lastingly attenuate the increased anxiety-like behavior by CRS. The anxiolytic effects of 10-min LS most likely resulted from its suppression of prefrontal glutamate release onto dmPFC→BLA PNs rather than LS per se because 10-min LS in GFP-only-expressing CRS mice failed to affect their

**Fig. 4 Chronic stress does not affect postsynaptic function in dmPFC synapses targeting BLA PN. a** Representative traces of EPSCs in dmPFC↔BLA PNs evoked by optogenetic activation of dmPFC inputs at −70 and +40 mV. Scale bar = 50 ms, 100 pA. Time points for determination of AMPA and NMDA receptor currents were indicated with arrows and dashed lines. **b** Summary plots of AMPA/NMDA ratio in **a**. Circles represent individual data points. Control, $n = 14$ neurons/5 mice; CRS, $n = 11$ neurons/4 mice. **c** Same as in **a** except that the data were from dmPFC→BLA PNs. Scale bar = 50 ms, 100 pA. **d** Summary plots of AMPA/NMDA ratio in **c**. Control, $n = 16$ neurons/6 mice; CRS, $n = 13$ neurons/4 mice. **e** Representative traces of light-evoked EPSCs in dmPFC↔BLA PNs when the holding potentials were increased from −80 to +40 mV at a step of 20 mV. Time points for determination of AMPA and NMDA receptor currents were indicated with arrows/dashed lines. Scale bar = 25 ms, 50 pA. **f, g** Current–voltage plots of the light-evoked AMPA (**f**) and NMDA (**g**) receptor currents in dmPFC↔BLA PNs. Current amplitudes were normalized to that recorded at −80 (**f**) or +40 mV (**g**). Control, $n = 14$ neurons/4 mice; CRS, $n = 12$ neurons/4 mice. **h** Same as in **e** except that the data were from dmPFC→BLA PNs. Scale bar = 25 ms, 50 pA. **i, j** Current–voltage plots of the light-evoked AMPA (**i**) and NMDA (**j**) receptor currents in dmPFC→BLA PNs. Current amplitudes were normalized to that recorded at −80 (**i**) or +40 mV (**j**). Control, $n = 13$ neurons/5 mice; CRS, $n = 14$ neurons/5 mice. **k** Representative traces of the asynchronized quantal EPSCs (qEPSCs) in dmPFC↔BLA PNs recorded when external $Ca^{2+}$ was replaced by same concentration of strontium ($Sr^{2+}$). The asterisks indicate detected qEPSC. Scale bar = 40 ms, 20 pA. **l, m** Summary plots of the frequency (**l**) and quantal size (**m**) of qEPSCs in dmPFC↔BLA PNs. Control, $n = 15$ neurons/5 mice; CRS, $n = 13$ neurons/4 mice. **n** Same as in **k** except that the data were from dmPFC→BLA PNs. Scale bar = 40 ms, 20 pA. **o, p** Summary plots of the frequency (**o**) and quantal size (**p**) of qEPSCs in dmPFC→BLA PNs. Control, $n = 17$ neurons/6 mice; CRS, $n = 15$ neurons/5 mice. Data are presented as mean ± SEM. Statistics are shown in Supplementary Table 1. n.s., not significant; **$p < 0.01$. Source data are provided as a Source Data file.

anxiety-like behavior (Fig. 7k–n, Supplementary Fig. 8g–i). In an additional experiment testing the immediate effect of 10-min LS on the anxiety-like behavior in CRS mice, we found that, relative to pre-LS, the behavioral parameters showed no changes during LS or 1 h post-LS (Supplementary Fig. 9), arguing against immediate effect of LS. Our findings thus highlight a critical role of dysregulated prefrontal control over dmPFC→BLA PNs in stress-induced increase of anxiety-like behavior (Fig. 8).

## Discussion
We here show that CRS dysregulates prefrontal control over BLA in a fashion heavily dependent on target cells' connectivity with PFC. Specifically, it shifts dmPFC-driven E–I balance in BLA PNs toward excitation but only in neurons receiving mono-directional afferents from dmPFC (dmPFC→BLA PNs) rather than those reciprocally connected with dmPFC (dmPFC↔BLA PNs). We link such specificity to the selective enhancement of prefrontal glutamate release onto the former population and show that using optogenetic stimulation of this pathway to normalize the increased glutamate release causes lasting reversal of stress-induced increase of anxiety-like behavior.

mPFC is a primary target of stress and engaged in multiple aspects of stress adversity on brain and behavior[41–45]. Chronic stress exposure can cause considerable architectural changes in the glutamatergic PNs in mPFC such as retraction of apical dendrites and loss of dendritic spines[46,47], resulting in dampened glutamatergic transmission onto them[48]. These changes have been thought as substrates for stress-induced mPFC dysfunction including its altered functional connectivity to amygdala and for the pathogenesis of stress-related psychiatric diseases[12,49,50]. Notably, recent evidence has begun to show that the architectural and functional remodeling of individual mPFC PNs by chronic stress varies with their specific project targets and molecular profiles[51,52]. As such, the BLA-projecting mPFC PNs are more resilient against the stress influences than those projecting to lateral entorhinal cortex, leading to intact dendritic arborization and unaltered spine density in this population upon chronic stress[51]. One would expect that such resilience might help to maintain the structural and functional integrity of the BLA-projecting mPFC neurons and thus not explain stress-induced impairment of prefrontal control over amygdala. Our finding that CRS dysregulates the dmPFC-to-BLA transmission may provide a plausible explanation. As generally known, the dmPFC projections to BLA are glutamatergic and presumably excitatory, the feeding of projections into local inhibitory network, however, yields strong feed-forward inhibition onto BLA PNs and results in

net inhibition of amygdala[5,21]. Here we observed that CRS markedly augmented the prefrontal excitatory transmission onto BLA PNs without affecting the local inhibitory tone, leading to a shift of dmPFC-driven E–I balance toward excitation. Such a shift is expected to weaken the prefrontal suppression of amygdala activity and output, thus exacerbating the development of stress-related emotional and behavioral disorders.

While our findings suggest an important role for augmented glutamatergic transmission in the altered prefrontal control of amygdala by CRS, a recent study by Wei et al, however, found that the defect was primarily associated with the diminished GABAergic transmission within BLA[53]. No sign of altered GABAergic transmission was observed in the current study. In considering this inconsistency, one should note that the current study and that by Wei et al. used different stressors (restraint versus unpredictable stress) with different timing (postnatal 7 versus 3 weeks). In fact, multiple lines of evidence have accumulated that the GABAergic signaling in BLA was more resilient to prolonged exposure to restraint stress than other stressors[54,55]. Moreover, the regulation of CRS on GABAergic versus glutamatergic system in amygdala is largely dependent on the timing of stress[56]. While the peri-pubertal CRS attenuates local GABAergic inhibition without altering glutamatergic transmission in amygdala, CRS during adulthood, on the other hand, preferentially enhances glutamatergic transmission but does not alter the GABAergic inhibition[56].

Previous studies have repetitively shown that CRS similarly affects the architecture of dendritic terminals of PNs from both hippocampus and prefrontal cortex, leading to dendritic hypotrophy and synaptic loss[46,57,58]. We found that the impacts of CRS on their output projections to BLA, however, were markedly different. While augmenting the glutamatergic transmission from dmPFC projection to BLA PNs, CRS had little influence on vHPC-to-BLA transmission, indicating region-specific regulation of BLA inputs. Although the exact reasons for the specificity are still vague, one potential interpretation is that the synapses made by vHPC onto BLA have far higher presynaptic glutamate release probability than those by mPFC[22], thus yielding a ceiling effect and preventing CRS from further increasing the Pr.

One important observation of the present study is that the CRS-induced, dmPFC-driven E–I imbalance in BLA PNs occurred in a manner heavily depending on the connectivity of the target cells. It shifted the E–I balance to excitation in dmPFC→BLA PNs but left it unaltered in dmPFC↔BLA ones. This shift is expected to provide more excitatory drive onto dmPFC→BLA PNs and thus likely contributes to the persistent

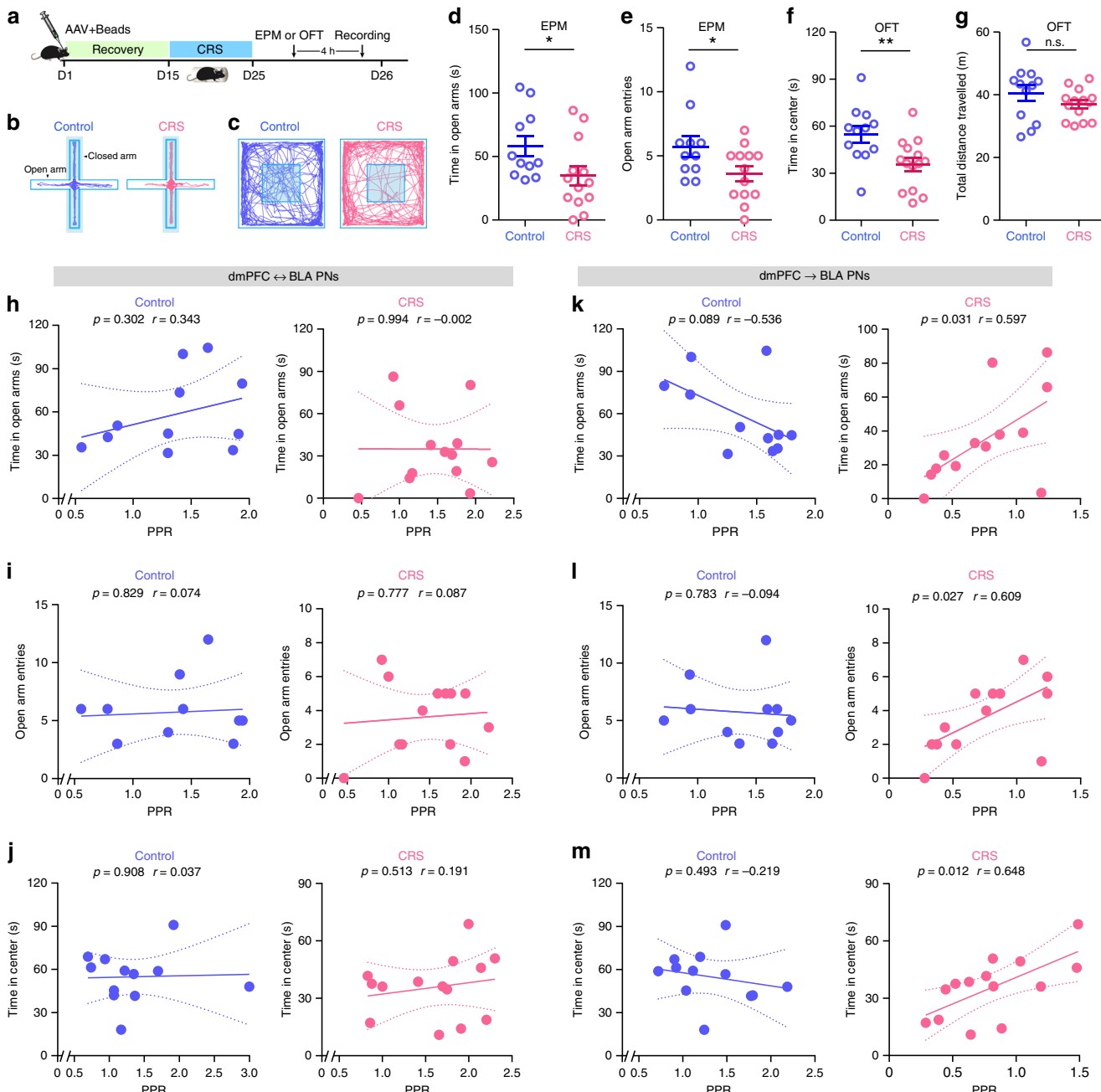

**Fig. 5 Correlation between chronic stress-induced increase of prefrontal glutamate release and anxiety-like behavior in mice. a** Schematic showing the experimental procedures. EPM elevated plus maze, OFT open field test. **b**, **c** Representative activity tracking in EPM (**b**) and OFT (**c**). **d**, **e** Time in open arms (**d**) and open-arm entries (**e**) during EPM. Control, $n = 11$ mice; CRS, $n = 13$ mice. **f**, **g** Time in center region (**f**) and total distance traveled (**g**) during OFT. Control, $n = 12$ mice; CRS, $n = 14$ mice. **h–j** Correlations between the PPR in dmPFC↔BLA PNs and open-arm time (**h**) and entry (**i**) in EPM and time in center region (**j**) in OFT in control and CRS mice. **k–m** Correlations between the PPR in dmPFC→BLA PNs and open-arm time (**k**) and entry (**l**) in EPM and time in center region (**m**) in OFT in control and CRS mice. For **d–g**, data are presented as mean ± SEM, and the statistics are shown in Supplementary Table 1. n.s., not significant; *$p < 0.05$, **$p < 0.01$. For **h–m**, linear regression analysis and Pearson's correlation were performed (two tailed). Source data are provided as a Source Data file.

activation of this population by CRS[29]. Moreover, the CRS effects on dmPFC-to-BLA transmission could be readily mimicked by chronic administration of CORT (Supplementary Figs. 5 and 6). Given the different influence of CORT treatment on dmPFC projection to dmPFC→BLA versus dmPFC↔BLA PNs, it is likely that the CORT signaling acts differently between the two projections. The dysregulated dmPFC transmission to dmPFC→BLA PNs, together with the increased dendritic spine density and neuronal excitability in this population[28,29], strongly implies that,

within the BLA microcircuits, the dmPFC→BLA PNs are more susceptible to the actions of chronic stress than their proximal dmPFC↔BLA neighbors and thus may have more prominent role in stress-related psychopathology.

We observed that the E–I imbalance in dmPFC projections to dmPFC→BLA PNs by CRS was due to the increased presynaptic glutamate releases onto this population. The increased glutamate release in dmPFC projection to dmPFC→BLA PNs, as reflected by the decreased PPR, was tightly correlated with the anxiety-like

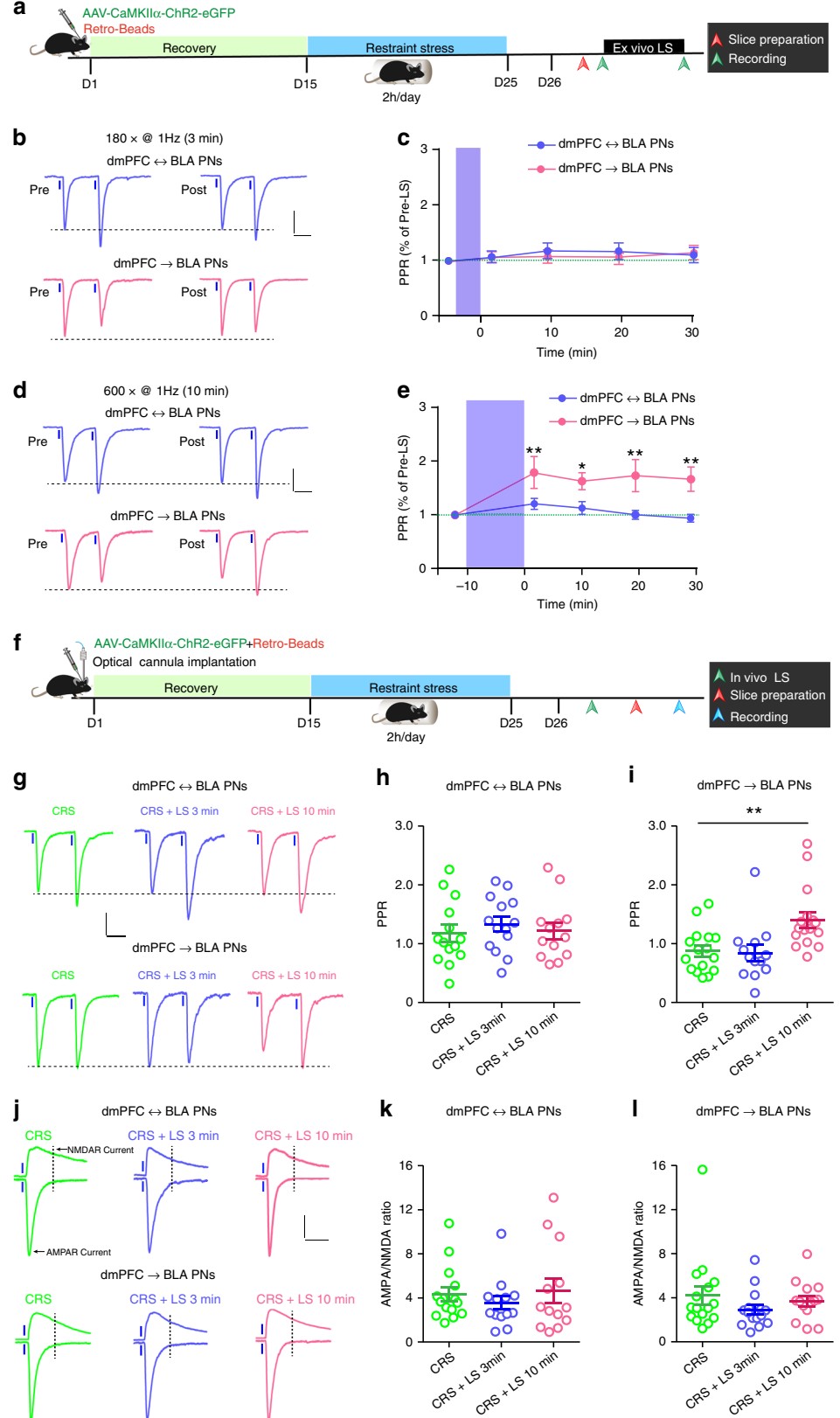

behavior in stressed mice, suggesting functional implications in the adversity by CRS. Of note is that such correlation is absent in unstressed mice, indicating that the role of dmPFC projections to BLA in tuning anxiety is not innate but develops during prolonged stress exposure. Likewise, in a task of adaptive adverse learning in monkeys, the prefrontal regulation of valence

encoding in BLA PNs was also reported to develop during training[59]. The exact reasons why the PPR is only related to the anxiety-like behavior in stressed but not in unstressed mice are still vague. It is expected that exposure to the behavioral apparatus testing anxiety will cause the mice, regardless of whether they are stressed or not, to release stress hormones, such as

**Fig. 6 Activation of dmPFC inputs reverses CRS-induced increase of glutamate release in dmPFC synapses terminating on dmPFC→BLA PNs.**
**a** Schematic showing the experimental procedures for ex vivo light stimulation (LS) and PPR recording. **b** Representative traces showing paired EPSCs in dmPFC↔BLA and dmPFC→BLA PNs of CRS mice prior and posterior to 3-min LS. Two light pulses were delivered at an interval of 100 ms. Scale bar = 50 ms, 50 pA. **c** Effect of 3-min LS on the PPR in the two PN populations of CRS mice. dmPFC↔BLA PNs, $n = 9$ neurons/6 mice; dmPFC→BLA PNs, $n = 10$ neurons/7 mice. **d** Same as in **b** except that the LS duration was extended to 10 min. **e** Effect of 10-min LS on the PPR in the two PN populations of CRS mice. dmPFC↔BLA PNs, $n = 9$ neurons/5 mice; dmPFC→BLA PNs, $n = 9$ neurons/6 mice. **f** Schematic showing experimental procedures for in vivo LS in CRS mice. **g** Representative traces showing paired EPSCs in dmPFC↔BLA PNs and dmPFC→BLA PNs from unstimulated CRS mice and mice receiving in vivo LS for 3 or 10 min. Scale bar = 50 ms, 50 pA. **h** Summary plots of PPR in dmPFC↔BLA PNs. CRS, $n = 14$ neurons/4 mice; CRS + 3 min LS, $n = 14$ neurons/5 mice; CRS + 10 min LS, $n = 13$ neurons/4 mice. **i** Summary plots of PPR in dmPFC→BLA PNs. CRS, $n = 16$ neurons/5 mice; CRS + 3 min LS, $n = 13$ neurons/4 mice; CRS + 10 min LS, $n = 16$ neurons/5 mice. **j** Representative traces of AMPA and NMDA receptor currents recorded in the dmPFC↔BLA and dmPFC→BLA PNs. Scale bar = 50 ms, 80 pA. **k** Summary plot of AMPA/NMDA ratio in dmPFC↔BLA PNs. CRS, $n = 15$ neurons/5 mice; CRS + 3 min LS, $n = 13$ neurons/4 mice; CRS + 10 min LS, $n = 13$ neurons/4 mice. **l** Summary plot of AMPA/NMDA ratio in dmPFC→BLA PNs. CRS, $n = 16$ neurons/5 mice; CRS + 3 min LS, $n = 15$ neurons/5 mice; CRS + 10 min LS, $n = 14$ neurons/4 mice. Data are presented as mean ± SEM. Statistics are shown in Supplementary Table 1. *$p < 0.05$, **$p < 0.01$. Source data are provided as a Source Data file.

CORT. However, it should be noted that CORT differently affects brain physiology and thus function in stressed versus unstressed mice. For example, while augmenting glutamatergic transmission in the amygdala of unstressed mice, CORT suppresses it in the stressed ones[60]. It is likely that the stress mediators (for example, CORT) may differently affect the PFC-to-BLA pathways in stressed versus unstressed mice, resulting in different engagement of this pathway in the anxiety-like behavior of these mice.

Synaptic dysfunction has been increasingly recognized as potential therapeutic target for stress-related mental diseases, such as depression[61]. Many of the psychoactive drugs have their therapeutic actions at least partially through reversing the synaptic dysfunction in mPFC[62]. We here observed that low-frequency (1 Hz) optogenetic stimulation of dmPFC-to-BLA pathway reversed the increased glutamate release onto dmPFC→BLA PNs and attenuated CRS-induced increase of anxiety-like behavior in mice, implying that targeting the synaptic pathology in this specific pathway might have translational value for treatment of stress-related neuropsychiatric diseases. Somewhat surprisingly, a previous study by Convington et al. reported that burst-like (100 Hz) optogenetic stimulation of the vmPFC had anti-depressive effect in chronically stressed mice[63]. The anti-stress influences by both high and low frequency of optogenetic stimulation of mPFC appear discrepant. However, at least two reasons suggest that it may not be the case. First, the high and low frequency of optical stimulation were delivered to ventral and dorsal parts of mPFC, respectively. Anatomical evidence has demonstrated that the two subregions of mPFC differ a lot in their connectivity with other brain regions such as amygdala[27,64], raising a possibility that they may be differently engaged in the emotional deficits by chronic stress. In support of this, recent studies have shown that the two mPFC subregions have different roles in tuning anxiety[65,66]. Second, unlike the selective suppression of dmPFC transmission to BLA by the low-frequency stimulation used in the present study, the high-frequency stimulation by Convington et al. recruited the whole network of vmPFC and its anti-stress effect, thus most likely reflecting consequence of altered vmPFC communication with all of its output regions. Different stimulation strategies should be considered when targeting the two mPFC subregions to treat stress-related psychiatric disorders.

While the current study has revealed target cell-dependent regulation of dmPFC-to-BLA pathway associated with chronic stress-induced increase of anxiety-like behavior, some important questions remain open. For example, although we have identified changes in presynaptic glutamate release as a key mechanism for differential adaptation of dmPFC synapses targeting dmPFC→ BLA versus dmPFC↔BLA PNs, what are the molecular machinery driving the different synaptic changes? Second, the BLA PNs have rich efferent targets, what are then the exact target(s) of the affected

dmPFC→BLA PNs by CRS? Answering these questions is expected to expand our understanding of how the coordination between amygdala and its upstream and downstream brain regions becomes maladapted leading to the development of stress-related psychiatric diseases.

## Methods

**Animal care**. Male C57BL/6J mice (5–10 weeks) were used for all experiments. The mice were initially purchased from Model Animal Research Center of Nanjing University and bred in animal facility of Nanchang University. The mice were housed in groups of 3–5 per cage with ad libitum access to food and water and maintained in a temperature (21–25 °C)- and humidity (40–60%)-controlled room with a light/dark cycle of 12 h (light on: 6:00 a.m.–6:00 p.m.). All experimental procedures were in accordance with the guidelines of the National Institutes of Health and approved by the Institutional Animal Care and Use Committee of Nanchang University.

**CRS and CORT treatment**. Mice were placed in a plastic air-accessible cylinder for 2 h (10:00–12:00) per day for 10 consecutive days (CRS). The size of the cylinder was similar to that of the animal, which made the animal almost immobile in the cylinder. The non-stressed controls were moved from the home cage to a test room and gently handled for 5 min before being returned to the holding room 2 h later. To chronically administer the mice with CORT, mice were allowed to have ad libitum access to water containing CORT (Sigma, St. Louis, MO, 0.1 mg ml$^{-1}$) for 10 consecutive days. The water was freshly prepared in opaque bottle to protect CORT from light. CORT was dissolved using ethanol to obtain a stock solution (10 mg ml$^{-1}$). The vehicle-treated mice were allowed to drink water containing 1% ethanol for 10 days[61]. To accurately control the consumed volume of CORT, commercial slow-release CORT pellets were used[35]. Briefly, a small incision was made on the side of the neck and CORT or placebo pellets (#G-111, Innovative Research of America, Sarasota, FL, USA) were placed in the incision. Electrophysiology experiments were performed 10 days after implantation of the pellets. The pellets are composed of a biodegradable matrix of cholesterol and cellulose and allow for continuous and sustained diffusion of CORT over a long period. Placebo pellets consisted of the same matrix without the active product.

**EPM test**. The EPM test was used to monitor anxiety-like behavior in mice. The maze apparatus consisted of two opposing open (35 cm × 6 cm) and two enclosed arms (35 cm × 6 cm) extending from a central platform (6 cm × 6 cm). The apparatus was raised 74 cm above the floor. During the test, mice were placed in the center square of the maze, facing an open arm, followed by a 10-min monitoring of their behavior. A video-tracking system (Med Associates Inc., Farifax, VT) was used to automatically track and analyze their entries into the open arms and the time they spent in the open arms. The apparatus was cleaned with 30% ethanol after each trial.

**Open field test**. The open field chamber was made of transparent plastic (50 cm × 50 cm) and a 25 cm × 25 cm center square was color marked. Individual mice were placed in the center of the chamber and their behavior was monitored for 10 min with an overhead video-tracking system (Med Associates Inc., Farifax, VT). The time mice spent in the center area, the total distance they traveled and their velocity were monitored throughout the experiment.

**Stereotaxic surgery and injections**. Five-to-6-week-old mice were used for stereotaxic injections[29]. Briefly, mice were anaesthetized with 2% pentobarbital sodium and placed in the stereotaxic frame (RWD, Shenzhen, China). For optogenetic manipulations of the dmPFC, vmPFC, or vHPC inputs to BLA inputs, the

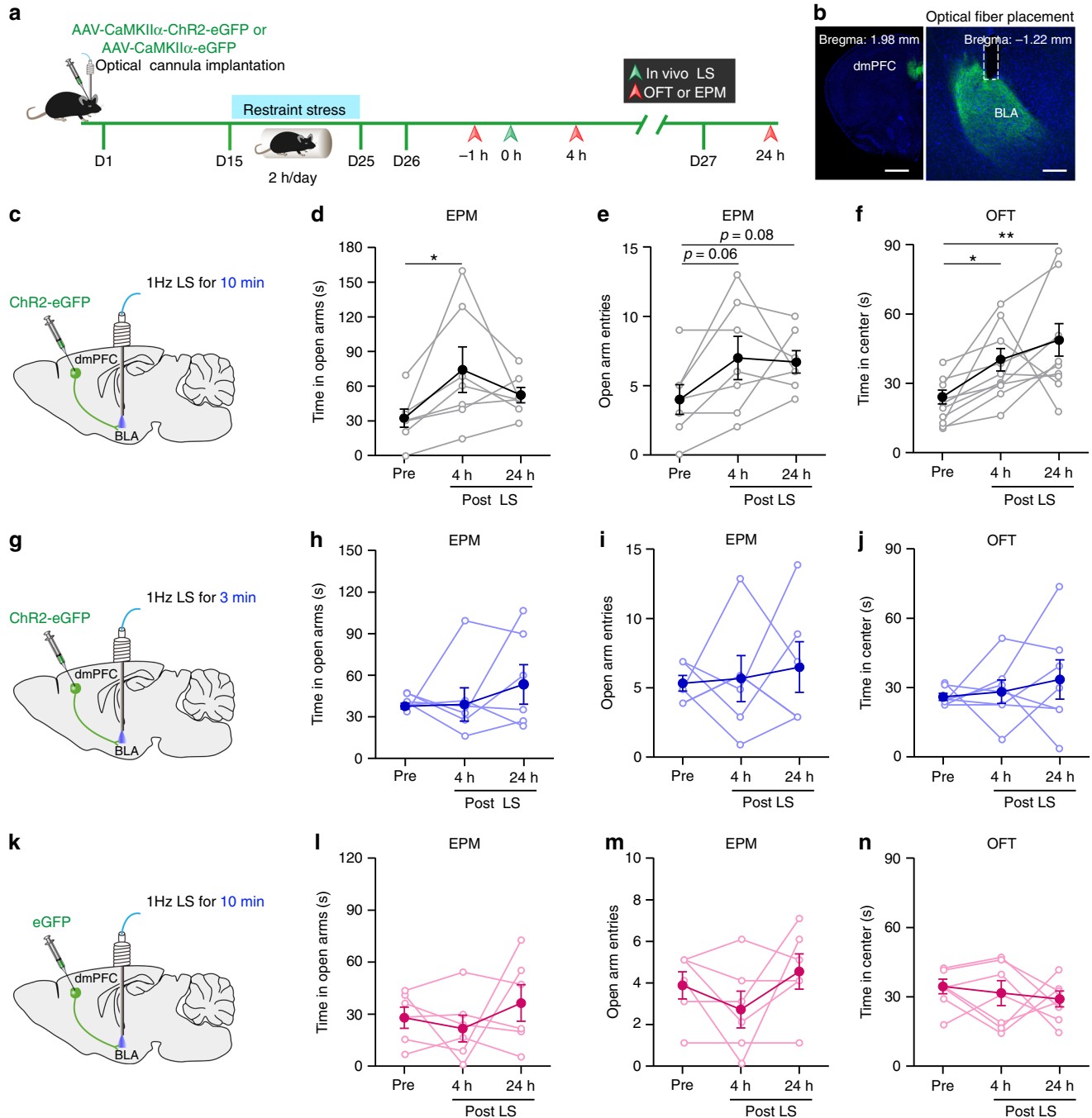

**Fig. 7 Activation of dmPFC-to-BLA pathway alleviates the increased anxiety-like behavior by chronic stress. a** Schematic showing the experimental procedures. **b** Representative image showing injection of ChR2-carrying AAV in dmPFC (left) and the cannula implantation onto BLA (right, dotted line). Scale bar: 500 (left) and 100 (right) µm. **c** Schematic illustration of injection of ChR2-carrying AAV in dmPFC and bilateral cannula implantation with optic fibers onto BLA. LS (1 Hz, 10 min) was delivered to BLA. **d**, **e** EPM open-arm time (**d**) and entries (**e**) measured pre- and post-LS. *n* = 7. **f** OFT time in center measured pre- and post-LS. *n* = 10. **g** Same as in **c** except that 3-min LS were delivered to CRS mice. **h**, **i** EPM open-arm time (**h**) and entries (**i**) measured pre- and post-LS. CRS mice, *n* = 6. Note that there were two data points in **i** being overlapped throughout the experiment. **j** OFT time in center measured pre- and post-LS. CRS mice, *n* = 7. **k** Same as in **c** except that the AAV vectors containing eGFP only were injected into dmPFC. **l**, **m** EPM open-arm time (**l**) and entries (**m**) measured pre- and post-LS. CRS mice, *n* = 6. **n** OFT time in center measured pre- and post-LS. CRS mice, *n* = 7. Data are presented as mean ± SEM. Statistics are shown in Supplementary Table 1. *$p < 0.05$. **$p < 0.01$. Source data are provided as a Source Data file.

anterogradely traveling AAV serotype 2/8 carrying ChR2 fused with eGFP under the control of the CaMKIIα promoter (AAV$_{2/8}$-CaMKIIα-hChR2(H134R)-eGFP, $4.72 \times 10^{12}$ infectious units mL$^{-1}$) (Obio Technology, Shanghai, China) were bilaterally injected into the dmPFC (0.35 µL per hemisphere; stereotaxic coordinates from bregma: anterior/posterior, 1.94 mm; medial/lateral, ±0.35 mm; dorsal/ventral, −2.5 mm), vmPFC (bregma coordinates: anterior/posterior, 1.94 mm; medial/lateral, ±0.35 mm; dorsal/ventral, −3.1 mm) or vHPC (bregma coordinates: anterior/posterior, −3.06 mm; medial/lateral, ±3.4 mm; dorsal/ventral, −3.8 mm).

When necessary, the red (Alexa Fluor 555) fluorescent retrogradely transported beads (RetroBeads, Lumafluor Inc., Durham, NC, USA) were co-injected with the ChR2-carrying AAV into the dmPFC, vmPFC, or vHPC to label the putative dmPFC↔BLA PNs, vmPFC↔BLA PNs, or vHPC↔BLA PNs in BLA. Injection was performed using glass micropipettes with their tip diameters of ~10–20 µm (pulled with the Narishige PC-10 puller, Japan) mounted on 10-µL Hamilton Microlitre syringe (Hamilton Co., Reno, NV, USA). The AAV and Retrobeads were delivered at a rate of 150 nL min$^{-1}$ using a stereotactic injector (QSI, Stoelting,

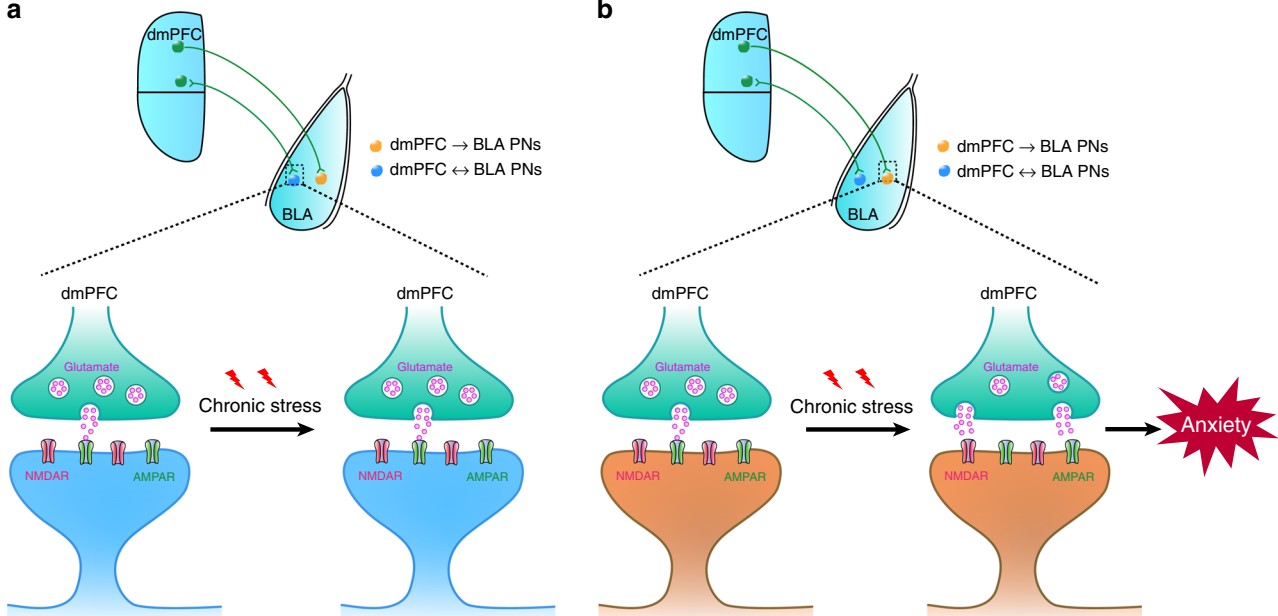

**Fig. 8 A working model for target cell-based dysregulation of dmPFC-to-BLA transmission associated with chronic stress-induced anxiety. a** Chronic stress does not affect prefrontal glutamate release onto BLA PNs that are reciprocally connected with dmPFC (dmPFC↔BLA PNs). **b** Chronic stress augments prefrontal glutamate release onto BLA PNs receiving mono-directional dmPFC inputs (dmPFC→BLA PNs), contributing to stress-induced increase of anxiety-like behavior in mice.

Wood Dale, IL, USA), with the pipette left in place for a further 10 min to allow diffusion.

**Histology and microscopy.** Mice were anesthetized with 2% pentobarbital sodium and transcardially perfused with 0.1 M ice-cold phosphate-buffered saline (PBS) followed by 4% paraformaldehyde (PFA). Brains were post-fixed overnight at 4 °C in 4% PFA and then cryopreserved in 30% sucrose. Coronal slices containing the dmPFC, vmPFC, BLA, or vHPC (30-µm thick) were cut using a freezing microtome (Leica CM 1950, Leica, Leica Microsystems, Wetzlar, Germany). Slices were then incubated with 4′,6-diamidino-2-phenylindole, a DNA-specific fluorescent probe for 5 min, rinsed 3× in PBS (3 × 5 min) followed by mounting with fluoromount aqueous mounting medium (Sigma-Aldrich, Saint Louis, Missouri, MO, USA). Confocal immunofluorescence images were taken by using a scanning laser microscope (Olympus FV1000, Tokyo, Japan).

**Amygdala slice preparation.** Mice were anesthetized with ether and decapitated, and the brains were rapidly removed and chilled in ice-cold, oxygenated (95% $O_2$ and 5% $CO_2$) artificial cerebrospinal fluid (ACSF) containing (in mM): 124 NaCl, 2.5 KCl, 2 $MgSO_4$, 2.5 $CaCl_2$, 1.25 $NaH_2PO_4$, 22 $NaHCO_3$, and 10 glucose. Coronal slices (320 µm) containing the BLA were cut using the VT1000S Vibratome (Leica Microsystems, Wetzlar, Germany). The slices were placed in warmed ACSF (34 °C) for 30 min and then maintained at room temperature for at least 1 h before recordings.

**Whole-cell patch clamp recordings.** Whole-cell patch clamp recordings were performed by using an infrared differential interference contrast microscope (BX51WI, Olympus, Tokyo, Japan) equipped with two automatic manipulators (Sutter Instrument Co., Novato, CA) and a highly sensitive CCD camera (IR-1000E, DAGE-MTI, Michigan, IN, USA). A single slice was transferred to the recording chamber and continuously perfused with oxygenated ACSF at a rate of ~2 mL min⁻¹. The temperature of ASCF was maintained at 29 ± 1 °C with an automatic temperature controller (TC-324B, Warner Instrument Co. Hamden, CT). Recording electrodes were made from filamented borosilicate glass capillary tubes (inner diameter, 0.84 µm) by using a horizontal pipette puller (P-97; Sutter Instrument Co., Novato, CA). The pipettes with resistance ranged from 3 to 6 MΩ were filled with intracellular solution containing (in mM): 130 Cs-methanesulfonate, 5 NaCl, 1 $MgCl_2$, 10 HEPES, 0.2 EGTA, 2 MgATP, 5 QX314, and 0.1 NaGTP, pH was adjusted to 7.30 with KOH. A junction potential of ~12 mV was uncorrected. In all, 100 µM PTX and 5 µM CGP52432 were routinely added into the bath solution unless otherwise mentioned.

To examine the monosynaptic nature of eEPSCs in BLA PNs following light activation of dmPFC inputs, slices were perfused with TTX (1 µM) to block the sodium channel blocker and TTX followed by addition of 4-aminopyridine (4-AP, 100 µM), a potassium channel blocker, to facilitate glutamate release from synaptic

terminals. The synaptic latencies of EPSC and IPSC were calculated as a time interval between the start of LS and the onset of current at holding potentials of −70 and 0 mV, respectively.

The input–output curves of synaptic responses were obtained from graded EPSCs and IPSCs induced by light pulse of increasing intensity (0.5, 1, 2, 5, and 10 µW).

In experiments measuring PPR of EPSCs, two light pulses (2 ms duration) with different interval (50, 100, 200, 500 ms) were delivered to BLA every 30 s. The light intensity was adjusted to achieve EPSCs with amplitude of 100–200 pA. The PPR was calculated as the ratio of the amplitude of the second EPSC over that of the first one.

In experiments measuring the decay of NMDA receptor current by MK-801 blockade, the recorded neurons were clamped at +40 mV in the presence of picrotoxin, CGP52432, and CNQX. The basal synaptic responses of BLA PNs to 0.1 Hz LS of dmPFC inputs were recorded for 3 min, followed by bath application of MK801 (20 µM) for 8 min without stimulation. The LS at 0.1 Hz was then resumed for 20 min.

To measure the AMPA/NMDA current ratio, cells were first clamped at −70 mV, and the AMPA receptor-mediated EPSCs of about 200–300 pA were recorded. The holding potential was then switched to +40 mV and the slowly decaying, NMDA receptor-mediated EPSCs were determined 50 ms after the peak of the current response when the contribution of the AMPAR current was minimal.

To evoke asynchronous synaptic responses, extracellular calcium was replaced with same concentration of strontium (2.5 mM). Baseline events were detected in the 600 ms preceding LS, and asynchronous events were detected during a 600-ms period beginning 30–50 ms after stimulation to eliminate synchronous synaptic responses.

Data were sampled at 10 kHz filtered at 2 kHz using the patch-clamp amplifier (EPC 10 USB, HEKA Instrument, Germany) circuitry and collected with the PATCHMASTER software (version 2.53). Series resistance (Rs) was in the range of 10–20 MΩ and monitored throughout the experiments. If Rs changed >20% during recording, the data were excluded. Offline data analysis was performed using Origin 8.5 (Microcal software, Northampton, MA, USA).

**Ex vivo optogenetics.** A light-emitting diode (LED) with 470 nm peak wavelength (M470L3, Thorlabs Inc., Newton, NJ, USA) was connected to an Master-8 Pulse Stimulator through the LED driver (LEDD1B, Thorlabs Inc.) to illuminate dmPFC-, vmPFC-, or vHPC-transfected fibers in BLA-containing brain slices in the recording chamber through a ×40 water-immersion objective lens (×40/NA0.8, LUMPlanFL, Olympus). Light intensity was measured with Optical Power Meter (PM100D power meter, Thorlabs Inc.).

**In vivo optogenetics.** Mice were implanted bilaterally with optical cannula in BLA (anterior/posterior, −1.28 mm; medial/lateral, ±3.2 mm; dorsal/ventral, −5.0 mm) immediately after AAV injection. Fifteen days after surgery, the mice were

subjected to CRS for 10 days, and their anxiety-like behavior were measured 24 h after the last restraint. The mice were moved to their home cage and allowed a 30-min acclimatization period, then the optical cannula was connected to the optical patch cable and 1 Hz pulses of blue light illumination (2 ms duration) for 3 or 10 min was delivered from a blue laser (470 nm, Newdoon Inc.) to the optical cannula through the optical patch cable and optical rotary joint. Behavioral testing was resumed 4 or 24 h post-stimulation.

**Statistics and reproducibility**. All data were presented as means ± SEM. The statistical analyses were performed by GraphPad Prism (GraphPad Software, Inc., San Diego, CA). The methods for statistical analysis and the sample size are described in figure legends. Data were analyzed using Student's $t$ test (for two groups), or one- or two-way analysis of variance with or without repeated measures, followed by post hoc comparisons with Bonferroni $t$ test. Corrections for multiple comparisons were made when necessary. The homoscedasticity and normality of the distributions were analyzed with Bartlett's and Kolmogorov–Smirnov tests, respectively. Pearson's correlation and linear regression analysis was performed to determine the correlation between PPR and anxiety-like behavior. The threshold for statistical significance was set at $p < 0.05$. The specific statistical tests used and the details of $p$ values for each experiment can be found in Supplementary Table 1. The histological experiments in Figs. 1b, j, q; 2b; and 7b and Supplementary Figs. 2b and 3b were repeated for three times and consistent results were observed.

**Reporting summary**. Further information on research design is available in the Nature Research Reporting Summary linked to this article.

## Data availability

The data supporting the findings of this study are available in the paper and Supplementary Information files or from the corresponding author upon reasonable request. A reporting summary for this article is available as a Supplementary Information file. The source data underlying Figs. 1f–h, m–o, t–v; 2f–i; 3b, d, f, h; 4b, d, f, g, i, j, l, m, o, p; 5d–m; 6c, e, h, i, k, l; 7d–f, h–j, l–n and Supplementary Figs. 1c, e; 2d–h; 3d–h; 4b, d, f, h; 5d–g; 6c–f; 7b, d; 8b, c, e, f, h, i; 9c–e, f–h, j–l, m–o are provided as a Source Data file.

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

## Acknowledgements

This work was supported by grants from the National Natural Science Foundation of China (Grant Nos. 81930032, 31970953, 81741759, 81601179, 31700916) and Natural Science Foundation of Jiangxi Province (20172BCB22005, 20192ACB20023, 20192ACB21024).

## Author contributions

W.-Z.L. and W.-H.Z. performed all the electrophysiological recordings and analyzed most of the data. Z.-H.Z. and X.-X.L. performed surgeries. W.-J.Y. and Y.H. raised the mice. J.-X.Z. performed histology studies. W.-Z.L, J.-X.Z., and S.-H.H. performed behavioral tests. W.-Z.L., W.-H.Z., X.-D.W., and B.-X.P. designed the experiments and wrote the manuscript with input from all authors.

## Competing interests

The authors declare no competing interests.
