## [Peer Review File · Nature Communications]

Reviewers' comments:

Reviewer #1 (Remarks to the Author):

This study by Liu et al probes the role of the prefrontal cortex-to-amygdala pathway in mediating stress induced behavioral changes (in particular anxiety). This is a technically sound novel study that identifies and characterizes the synaptic function of two unique populations of cells in principle neurons in amygdala: one that projects back to PFC, and one that does not. Through a series of elegant experiments, the authors clearly establish that changes in presynaptic release in the PFC->AMY pathway, but not within the PFC <-> AMY pathway occur in response to stress and subsequently mediate increased anxiety. I found it a strength that the authors explored this phenomenon in multiple stress paradigms (chronic restraint and chronic CORT), strengthening their argument that the phenomenon generalized. Finally, the authors also use an optogenetic approach to modify synaptic strength in this pathway and demonstrate that they can regulate behavior in the stress model. This work would be of interest to the Nature Communications audience, but I do have several critiques that the authors should address prior to publication.

1) The authors frame their findings as being due to changes in the E/I ratio in the PFC->AMY circuit. To make this case, they show that there are no changes in the evoked inhibitory currents. But there is a problem here. They only establish that the evoked activity in the E pathway increases while the evoked activity in the I pathway does not. The authors don't actually demonstrate that the activity in the E pathway is increased in the endogenous setting. For example, it's totally possible that a decrease in the firing of neurons paired with an increase in release yields no function change in the E pathway. My point here is that if the authors want to make the case about changes in the E/I balance, they should either show that the presynaptic neurons in the PFC-AMY pathway don't change or increase their firing rates, or 2) they should show that the postsynaptic neurons increase their firing rates, which the postsynaptic neurons in the PFC<->AMY pathway do not.

2) It would certainly strengthen the authors' claims if they demonstrated that selectively potentiating the PFC->AMY pathway was sufficient to drive anxiety in non-stressed animals.

Minor

1) Why VHPC -> BLA as a pathway. Aren't there reciprocal projections in this pathway. Are there 2 populations of cells in AMY that receive projections from VHPC (one that senses projections back and ones that don't). The authors should clarify this since it would directly impact the interpretation of these control experiments.

2) The authors state "the vast majority of BLA PNs are innervated by dmPFC." They should actually

provide some quantification here. 90%, 95%, 99%.

3) Line 237 should read dmPFC -> BLA not BLA->dmPFC.

4) Line 55, small fraction. 10-20% is not a small fraction. Especially in a feed forward non-linear system. The FSIs make 100s of synaptic connections with PNs. Just stating the percentages is totally fine.

5) the authors use the word defective to describe PFC->AMY. The word 'altered' would be better. Defective can imply a lack of function, when in this case, the pathway is hyperfunctional.

6) What statistical tests did the authors use for the correlations in Fig 5. Parametric, non-parametric?

7) Lines 357-9: The authors state their results "highlight an important role for CORT signaling." I disagree with this wholeheartedly. Their results show a convergent mechanism across two stress paradigms. If they wish to show the importance of CORT signaling, they should do necessary and sufficient experiments by directly manipulating the CORT signaling pathways while observing the impact on their measured phenomenon (synaptic strength).

Reviewer #2 (Remarks to the Author):

Liu and collaborators performed a very elegant study and investigated the impact of chronic stress exposure on prefrontal cortex (dmPFC) to basolateral amygdala (BLA) neural transmission and consecutive pathological behaviors such as anxiety-like behaviors. Using a combination of electrophysiological, optogenetic and behavior and a rodent model for anxiety (i.e. chronic restraint stress, CRS), the authors established that exposure to chronic stress alters dmPFC-BLA transmission in a selective manner. Acknowledging the large heterogeneity of BLA principal neurons (PN), the authors delved deeper and identified that CRS alters dmPFC->BLA principal neuron's (PN) EPSCs without impacting dmPFC<->BLA PN EPSCs and ventral hippocampus to BLA PN EPSCs. Further, the authors expand on their findings and confirm the selective impact of stress, using corticosterone treatment, on dmPFC->BLA PN transmission. Finally the authors combined optogenetic techniques and behavioral approaches to establish a causal link between alteration of dmPFC->BLA PN transmission and anxiety-like behaviors.

Overall the manuscript is clear and the experiments are well performed. The current data set and interpretations are very interesting and will provide useful information to the field of neuroscience and psychiatric disorders. Our current enthusiasm for this manuscript would increase even more if the authors addressed the outlined points below that I hope will improve this very interesting study.

Major Comments:

- The authors used a low frequency (1Hz) optogenetic stimulation of the dmPFC-to-BLA pathway to

reverse CRS-induced anxiety-like effects. The authors also mentioned a previous investigation (Covington et al 2010) using high frequency (100Hz) stimulation to rescue acutely depressive-like behavior without affecting anxiety-like behavior. This is a potential interesting result as the authors here observed a sustained anxiolytic effect. The authors should investigate the impact of acute and simultaneous optogenetic stimulation (1Hz) and also assess how long the observed anxiolytic effect persists. These results will provide very useful information regarding the contribution of PFC-BLA transmission in the expression of both depressive- and anxiety-like behaviors.

- The authors performed a multi-technique approach builds a strong foundation for the author's hypotheses and investigation that could be even more convincing if the sample size (i.e. mice number) was increased from 2-3mice to 4-5 mice.

Minor Comments:

- The authors should provide power analyses to determine the sample size required to observe significant effects in their study. They also should provide the statistical analyses to attest the normality and homoscedasticity of the data sets.

- The authors should provide the titer of the virus used in their study.

- Few sentences could be rewritten to ease the understanding of the authors interpretation, e.g. L177-178, L222-226, L249-251, L381-385.

Reviewer #3 (Remarks to the Author):

Liu and colleagues present data that mono-directional inputs from dmPFC to BLA rather than those reciprocally connected are dysregulated following chronic stress and lead to aberrant anxiety-like behavior. They have used slice physiology and optogenetics to support this finding.

The manuscript presents intriguing data in a rather convincing way. I have some concerns, however, with regard to completeness of story (i.e., selection of regions) as well as a few details related to behavior.

Most importantly, we know that the PFC exerts control over the amygdala to regulate anxiety and fear, and that "PFC" here includes infralimbic/ventromedial PFC in addition to prelimbic/dmPFC subregions. Furthermore, evidence has also shown that trauma can alter vmPFC-amygdala connectivity and/or affect glutamate transmission in vmPFC neurons. As such, it is unclear why the authors did not consider vmPFC projections in addition to dmPFCBLA since there is such extensive evidence that IL/vmPFC  amygdala regulates anxiety. This should be addressed at the very least. Including vmPFC experiments that parallel dmPFC would be a valuable addition to the manuscript.

The authors administered chronic CORT treatment via drinking water. Is there a reason this was done as opposed to controlling volume of CORT administered across animals? It seems likely that there would be a direct correlation between volume of CORT consumed and magnitude of dmPFC-evoked EPSCs. Was this the case? It would be interesting to note whether behavior/ physiology measured correlated in any way to the volume of CORT consumed (or if this wasn't done, why that was the case).

While this is briefly addressed in the discussion, a potential issue with the EPM data (fig 5) is the lack of correlation between anxiety-like behavior and PPR in non-stressed animals. Regardless of stress history, performance on the EPM is attributed to anxiety-like behavior in general (and exposure to the EPM increases CORT in animals regardless of whether they enter open arms or not). Authors state that dmPFC glutamate may mediate anxiety only if an animal has a history of CRS. What would mediate anxiety then in the non-stressed animal that is unrelated to glutamate, and related, since this effect is stressor specific, what does that mean for translational value? This needs to be addressed.

Response to the Reviewers' comments

We thank all the three reviewers for the many valuable comments they made, which we have incorporated to the benefit of the manuscript.

Reviewer #1

This study by Liu et al probes the role of the prefrontal cortex-to-amygdala pathway in mediating stress induced behavioral changes (in particular anxiety). This is a technically sound novel study that identifies and characterizes the synaptic function of two unique populations of cells in principle neurons in amygdala: one that projects back to PFC, and one that does not. Through a series of elegant experiments, the authors clearly establish that changes in presynaptic release in the PFC->AMY pathway, but not within the PFC <-> AMY pathway occur in response to stress and subsequently mediate increased anxiety. I found it a strength that the authors explored this phenomenon in multiple stress paradigms (chronic restraint and chronic CORT), strengthening their argument that the phenomenon generalized. Finally, the authors also use an optogenetic approach to modify synaptic strength in this pathway and demonstrate that they can regulate behavior in the stress model. This work would be of interest to the Nature Communications audience, but I do have several critiques that the authors should address prior to publication.

1. The authors frame their findings as being due to changes in the E/I ratio in the PFC->AMY circuit. To make this case, they show that there were no changes in the evoked inhibitory currents. But there is a problem here. They only establish that the evoked activity in the E pathway increases while the evoked activity in the I pathway does not. The authors don't actually demonstrate that the activity in the E pathway is increased in the endogenous setting. For example, it's totally possible that a decrease in the firing of neurons paired with increase in release yields no function change in the E pathway. 1) My point here is that if the authors want to make the case about

changes in the E/I balance, they should either show that the presynaptic neurons in the PFC-AMY pathway don't change or increase their firing rates, or 2) they should show that the post synaptic neurons increase their firing rates, which the postsynaptic neurons in the PFC->AMY pathway do not.

We thank the reviewer for identifying this issue and have made attempts to address the issue. For the 1st possibility, in one of our ongoing projects [REDACTED]

[REDACTED] we have observed that the AMY-projecting PNs showed increased excitability upon CRS. [REDACTED]

[REDACTED] For the 2nd possibility, in our recent Biological Psychiatry paper (Zhang et al, 2019, 85(10):812-828. Fig.3c-f), we have actually shown that CRS significantly increased the firing of dmPFC->BLA but not dmPFC<->BLA PNs. Given these, we speculate that the increased activity in E pathway most likely occur in endogenous settings.

2. It would certainly strengthen the authors claims if they demonstrated that selectively potentiating the PFC->AMY pathway was sufficient to drive anxiety in non-stressed animals.

We agree with the reviewer's comment that the claim would be strengthened if selectively potentiating the PFC inputs to dmPFC->Amy PNs was sufficient to drive

anxiety-like behavior in unstressed mice. Since no approach is available allowing us to selectively potentiate dmPFC inputs to dmPFC→Amy PNs (but leave those to dmPFC↔Amy PNs unaltered), we first made attempt to screen for the potential approaches. We tested two protocols which were successfully used to augment glutamatergic transmission and alter behavior. The 1st is high-frequency light stimuli (100 Hz, 100 stimuli, repeated 6 times with an interval of 20 s, adopted from Zhou et al, *Science*, 2017, 357: 162-168) and the 2nd is middle-frequency light stimuli (10 Hz, 100 stimuli, repeated 6 times with an interval of 20 s, with modification from Liu et al, *J. Neurosci.* 2016, 36: 7897-7910). We found that unlike the low-frequency light stimuli (Fig. 6b-d) selectively regulating dmPFC outputs to different BLA PNs in stressed mice, the above two protocols decreased the PPR similarly in both pathways of the unstressed controls. Thus, we failed to find protocols to selectively manipulate the dmPFC inputs to dmPFC→BLA PNs in unstressed mice, and, because of this, we have to pause the experiment. We are sorry for this.

a Summary plots of paired pulse ratio (PPR) in dmPFC↔BLA PNs from unstimulated control mice and mice receiving in vivo LS (100Hz). **p < 0.01. **b** Same as in (a) except that the data were from dmPFC→BLA PNs. *p < 0.05. **c-d** Same as in (a-b) except that the frequency was set at 10 Hz. **p < 0.01.

3. Why vHPC → BLA as a pathway. Aren't there reciprocal projections in this pathway? Are there 2 populations of cells in AMY that receive projections from vHPC (one that sense projections back and ones that don't)? The authors should clarify this since it would directly impact the interpretation of these control experiments.

We selected vHPC→BLA as a control pathway in that vHPC and mPFC are believed to transmit different components of information to BLA (vHPC: spatial information; mPFC: executive/cognitive information). Actually, there are reciprocal projections in

vHPC→BLA pathway: some BLA PNs are reciprocally connected with vHPC while the others only receive vHPC inputs. During the revision, we also tested the CRS effects on vHPC inputs to vHPC→BLA vs vHPC↔BLA PNs and found that consistent with its negligible influence on vHPC transmission to BLA PNs as a whole, it also unaffected the transmission to the two subpopulations. The data were shown in Fig. S3. Per the request of other reviewers, we also used vmPFC→BLA pathway as an additional control (Fig. 1i-o, Fig. S2).

4. The authors state "the vast majority of BLA PNs are innervated by dmPFC." They should actually provide some quantification here. 90%, 95%, 99%?

Based on the finding from Vadims' lab that 215 out of the 215 recorded neurons are activated by dmPFC inputs (Cho et al, Neuron, 2013, 80: 1491-1507), it appears that virtually all BLA PNs are innervated by dmPFC. We have added this in Line 134.

5. Line 237 should read dmPFC -> BLA not BLA->dmPFC.

We have corrected the error.

6. Line 55, small fraction. 10-20% is not a small fraction. Especially in a feed forward non-linear system. The FSIs make 100s of synaptic connections with PNs. Just stating the percentages is totally fine.

We have removed the words "small fraction".

7. The authors use the word defective to describe PFC->AMY. The word 'altered' would be better. Defective can imply a lack of function, when in this case, the pathway is hyperfunctional.

We have replaced the word "defective" with "altered".

8. What statistical tests did the authors use for the correlations in Fig 5. Parametric, non-parametric?

To determine the correlation between PPR and anxiety-like behavior (Fig. 5), we performed linear regression analysis and Pearson's correlation (parametric tests). We have stated this in the statistical analyses section.

9. Lines 357-9: The authors state their results "highlight an important role for CORT

signaling." I disagree with this wholeheartedly. There results show a convergent mechanism across two stress paradigms. If they wish to who the importance of CORT signaling, they should do necessity and sufficient experiments by directly manipulating the CORT signaling pathways while observing the impact on their measured phenomenon (synaptic strength).

We agree that the statement is not conclusive and we have removed it in the revised version.

Reviewer 2

Liu and collaborators performed a very elegant study and investigated the impact of chronic stress exposure on prefrontal cortex (dmPFC) to basolateral amygdala (BLA) neural transmission and consecutive pathological behaviors such as anxiety-like behaviors. Using a combination of electrophysiological, optogenetic and behavior and a rodent model for anxiety (i.e. chronic restrain stress, CRS), the authors established that exposure to chronic stress alters dmPFC-BLA transmission in a selective manner. Acknowledging the large heterogeneity of BLA principal neurons (PN), the authors delved deeper and identified that CRS alters dmPFC->BLA principal neuron's (PN) EPSCs without impacting dmPFC<->BLA PN EPSCs and ventral hippocampus to BLA PN EPSCs. Further, the authors expand on their findings and confirm the selective impact of stress, using corticosterone treatment, on dmPFC->BLA PN transmission. Finally the authors combined optogenetic techniques and behavioral approaches to establish a causal link between alteration of dmPFC->BLA PN transmission and anxiety-like behaviors. Overall the manuscript is clear and the experiments are well performed. The current data set and interpretations are very interesting and will provide useful information to the field of neuroscience and psychiatric disorders. Our current enthusiasm for this manuscript would increase even more if the authors addressed the outlined points below that I hope will improve this very interesting study.

1. The authors used a low frequency (1Hz) optogenetic stimulation of the

dmPFC-to-BLA pathway to reverse CRS-induced anxiety-like effects. The authors also mentioned a previous investigation (Covington et al 2010) using high frequency (100Hz) stimulation to rescue acutely depressive-like behavior without affecting anxiety-like behavior. This is a potential interesting result as the authors here observed a sustained anxiolytic effect. The authors should investigate the impact of acute and simultaneous optogenetic stimulation (1Hz) and also assess how long the observed anxiolytic effect persists. These results will provide very useful information regarding the contribution of PFC-BLA transmission in the expression of both depressive- and anxiety-like behaviors.

We have followed the reviewer's suggestion and assessed the acute and simultaneous effect of optogenetic stimulation (1 Hz). The results showed that relative to pre-LS, the anxiety-like behavior was unaltered during LS or 1h post LS (Fig. S9), arguing against acute anxiolytic effect of LS.

To assess how long the anxiolytic effect of LS could persist, we compared the anxiety-like behaviors pre-LS and 1 week post-LS in CRS mice injected either with ChR2- or GFP-carrying AAV. As shown below, although the anxiolytic-like phenotype was still evident in ChR2-injected mice 1 week post LS, it was also observed in mice injected with control virus. These results suggest that the mice have the ability to get recovered from CRS 1 week post-stress (at least in terms of the anxiety-like behavior) even in the absence of stimulation of dmPFC-to-BLA pathway, which prevent us from exploring the long term influence of LS using the current CRS protocol.

a Elevated plus maze open arm time measured pre-LS and 1 week post-LS. **b** EPM open arm entries measured pre-LS and 1 week post-LS. **c** OFT time in center measured pre-LS and 1 week post-LS. * $p < 0.05$.

2. The authors performed a multi-technique approach builds a strong foundation for the author’s hypotheses and investigation that could be even more convincing if the sample size (i.e. mice number) was increased from 2-3mice to 4-5 mice.

We have followed the reviewer’s suggestion and increased the mice size from 2-3 to 4-5.

3. The authors should provide power analyses to determine the sample size required to observe significant effects in their study. They also should provide the statistical analyses to attest the normality and homoscedasticity of the data sets.

Upon submission of the initial manuscript, we have claimed in the Reporting Summary that no statistical methods were used to predetermine the sample sizes, but the sizes were based on our previous studies performing similar experiments (Liu et al, Biol. Psychiatry, 2017, 81: 990-1002, Zhang et al, Biol. Psychiatry, 2019, 85: 189-201.) and convention in the field. Here, we compared the actual sizes and the sizes determined with reference to “sample size determination” (Dell et al, ILAR J, 2002, 43(4): 207-213) and found that in most experiments, the sample sizes were sufficient. For some in which the sizes are relatively low, we have increased them correspondingly (for example, following the comment 2). We used Bartlett’s test and

K-S test to attest the normality and homoscedasticity of the data sets respectively.
We have stated this in the statistical analyses section.

4. The authors should provide the titer of the virus used in their study.

The titer of the virus has been provided.

5. Few sentences could be rewritten to ease the understanding of the authors' interpretation, e.g. L177-178, L222-226, L249-251, L381-385.

We have followed the reviewer's suggestions and rewritten the above sentences to ease the understanding.

Reviewer #3:

Liu and colleagues present data that mono-directional inputs from dmPFC to BLA rather than those reciprocally connected are dysregulated following chronic stress and lead to aberrant anxiety-like behavior. They have used slice physiology and optogenetics to support this finding. The manuscript presents intriguing data in a rather convincing way.

1. I have some concerns, however, with regard to completeness of story (i.e., selection of regions) as well as a few details related to behavior. Most importantly, we know that the PFC exerts control over the amygdala to regulate anxiety and fear, and that "PFC" here includes infralimbic/ventromedial PFC in addition to prelimbic/dmPFC subregions. Furthermore, evidence has also shown that trauma can alter vmPFC-amygdala connectivity and/or affect glutamate transmission in vmPFC neurons. As such, it is unclear why the authors did not consider vmPFC projections in addition to dmPFCBLA since there is such extensive evidence that IL/vmPFC  amygdala regulates anxiety. This should be addressed at the very least. Including vmPFC experiments that parallel dmPFC would be a valuable addition to the manuscript.

We agree with the reviewer that including vmPFC experiments would be a valuable

addition to the manuscript. We also noted this during peer review of the paper (and also follow the reviewer's suggestion) and started to test the CRS effect on vmPFC inputs to BLA PNs. The results showed that in sharp contrast to its dramatic influence on dmPFC pathway, CRS had little influence on vmPFC projection to BLA PNs, either when the cells were treated as a whole or separated into vmPFC→BLA or vmPFC↔BLA PNs (Fig. 1i-o, Fig. S2). A recent study also revealed different adaptation of dmPFC- vs vmPFC-to BLA pathways in response to chronic alcohol administration (McGinnis et al, eNeuro, 2019), suggesting that the two inputs may use different strategies to cope with external challenges.

2. The authors administered chronic CORT treatment via drinking water. Is there a reason this was done as opposed to controlling volume of CORT administered across animals? It seems likely that there would be a direct correlation between volume of CORT consumed and magnitude of dmPFC-evoked EPSCs. Was this the case? It would be interesting to note whether behavior/ physiology measured correlated in any way to the volume of CORT consumed (or if this wasn't done, why that was the case).

We added CORT to the drinking water to increase CORT levels in mice with reference to the previous works from our and other's lab (Liu et al, Mol Brain, 2014; David et al, Neuron, 2009, 62: 479-493). We have confirmed that this approach is effective in increasing the CORT level in mice (Liu et al, Mol Brain, 2014). We agree with the reviewer that there would be a direct correlation between the volume of CORT consumed and magnitude of dmPFC-evoked EPSCs. During the revision, we repeated the tests by controlling the CORT volume via commercial slow-release CORT pellets or placebo (Adhikari et al, Nature, 2015, 527: 179-185). The pellets contain 10 mg of corticosterone and can be used for as long as 21 days. The results showed that similar to administering CORT treatment via drinking water, controlling CORT volume also augmented the excitatory transmission from dmPFC to dmPFC→BLA but not dmPFC↔BLA PNs (Fig. S6, Line173-178).

3. While this is briefly addressed in the discussion, a potential issue with the EPM

data (fig 5) is the lack of correlation between anxiety-like behavior and PPR in non-stressed animals. Regardless of stress history, performance on the EPM is attributed to anxiety-like behavior in general (and exposure to the EPM increases CORT in animals regardless of whether they enter open arms or not). Authors state that dmPFC glutamate may mediate anxiety only if an animal has a history of CRS. What would mediate anxiety then in the non-stressed animal that is unrelated to glutamate, and related, since this effect is stressor specific, what does that mean for translational value? This needs to be addressed.

The exact reasons why PPR is only related to anxiety-like behavior in stressed but not unstressed mice are still unclear. As the reviewer stated, exposure to the EPM would increase CORT in both unstressed and unstressed mice. However, it should be noted that the CORT effects on brain physiology and thus function are different in the two mice groups. For example, while augmenting glutamatergic transmission in the amygdala of unstressed mice, CORT suppresses it in the stressed ones (Karst et al., PNAS, 2010, 107: 14449-14454). It is likely that the stress mediators (for example, CORT) may differently affect the PFC-to-BLA pathways in stressed versus unstressed mice, resulting in different engagement of this pathway in the anxiety-like behavior of these mice. We have added this to the Discussion (Line 384-393).

In terms of the brain mechanisms underlying anxiety in unstressed controls, it has been shown that pathways such as the BLA-to-vHPC and BLA-to-BNST pathways are engaged in regulating anxiety-like behavior in unstressed mice (Calhoun GG and Tye KM. Nat Neurosci, 2015, 18: 1394-1404). Thus, although our findings revealed no clear correlation between the glutamate release in dmPFC-to-BLA pathway and anxiety-like behaviors in unstressed mice, the correlation may exist for the glutamate in these pathways.

We have discussed the potential translational value of our finding in the Discussion (Line 397-418)

****REVIEWERS' COMMENTS:**

Reviewer #1 (Remarks to the Author):

The authors have addressed all of my concerns.

Reviewer #2 (Remarks to the Author):

The authors have been very responsive to the reviewer's comments and suggestions. We have no further comments or concerns.

Reviewer #3 (Remarks to the Author):

The authors have addressed my concerns satisfactorily, and have improved the quality and scope of their manuscript based on all reviewers' comments.